# SARS-CoV-2 Nsp14 mediates the effects of viral infection on the host cell transcriptome

Michela Zaffagni, Jenna M Harris, Ines L Patop, Nagarjuna Reddy Pamudurti, Sinead Nguyen, Sebastian Kadener*

Department of Biology, Brandeis University, Waltham, United States

**Abstract** Viral infection involves complex set of events orchestrated by multiple viral proteins. To identify functions of SARS-CoV-2 proteins, we performed transcriptomic analyses of cells expressing individual viral proteins. Expression of Nsp14, a protein involved in viral RNA replication, provoked a dramatic remodeling of the transcriptome that strongly resembled that observed following SARS-CoV-2 infection. Moreover, Nsp14 expression altered the splicing of more than 1000 genes and resulted in a dramatic increase in the number of circRNAs, which are linked to innate immunity. These effects were independent of the Nsp14 exonuclease activity and required the N7-guanine-methyltransferase domain of the protein. Activation of the NFkB pathway and increased expression of *CXCL8* occurred early upon Nsp14 expression. We identified IMPDH2, which catalyzes the rate-limiting step of guanine nucleotides biosynthesis, as a key mediator of these effects. Nsp14 expression caused an increase in GTP cellular levels, and the effect of Nsp14 was strongly decreased in the presence of IMPDH2 inhibitors. Together, our data demonstrate an unknown role for Nsp14 with implications for therapy.

## Editor's evaluation

The paper shows that expression of the SARS-CoV-2 Nsp14 protein, which is involved in viral RNA replication, provokes a transcriptional profile that strongly resembles that observed following SARS-CoV-2 infection. Moreover, Nsp14 expression alters the splicing of many genes, increases the number of circRNAs, and activates the NFkB pathway. This surprising observation gives new insight into the biology of SARS-CoV-2 and may have implications for therapy.

*For correspondence: skadener@brandeis.edu

Competing interest: The authors declare that no competing interests exist.

## Introduction

Severe acute respiratory syndrome coronavirus 2 (SARS-CoV-2) is the virus responsible for the COVID-19 pandemic that began in 2019. As of early February 2022, COVID-19 has caused 5.7 million deaths worldwide. Coronaviruses are enveloped, relatively small (60–140 nm diameter), positive-stranded RNA viruses belonging to the Coronaviridae family. They derive their name from the crown-like appearance (corona means crown in Latin) that results from the spike glycoproteins in their envelope (*V'kovski et al., 2021*). The SARS-CoV-2 RNA genome is 30 kb long and has 14 open reading frames (ORFs) that encode 29 proteins (16 non-structural proteins, 4 structural proteins, and 9 accessory factors, *Kim et al., 2020*). During the first step of the viral infection, the spike glycoprotein on the viral envelope mediates attachment and fusion with the cellular membrane. Once the virus is in the cytoplasm of the host cells, the host ribosome machinery is recruited for the synthesis of viral proteins. Non-structural proteins are required for viral genome replication and are generated by proteolytic cleavage of the polyprotein encoded by ORF1a and ORF1ab. Once the viral genome is replicated,

**eLife digest** Viruses are parasites, relying on the cells they infect to make more of themselves. In doing so they change how an infected cell turns its genes on and off, forcing it to build new virus particles and turning off the immune surveillance that would allow the body to intervene. This is how SARS-CoV-2, the virus that causes COVID, survives with a genome that carries instructions to make just 29 proteins.

One of these proteins, known as Nsp14, is involved in both virus reproduction and immune escape. Previous work has shown that it interacts with IMPDH2, the cellular enzyme that controls the production of the building blocks of the genetic code. The impact of this interaction is not clear.

To find out more, Zaffagni et al. introduced 26 of the SARS-CoV-2 proteins into human cells one at a time. Nsp14 had the most dramatic effect, dialing around 4,000 genes up or down and changing how the cell interprets over 1,000 genes. Despite being just one protein, it mimicked the genetic changes seen during real SARS-CoV-2 infection. Blocking IMPDH2 partially reversed the effects, which suggests that the interaction of Nsp14 with the enzyme might be responsible for the effects of SARS-CoV-2 on the genes of the cell.

Understanding how viral proteins affect cells can explain what happens during infection. This could lead to the discovery of new treatments designed to counteract the effects of the virus. Further work could investigate whether interfering with Nsp14 helps cells to overcome infection.

---

virions are assembled in the host endoplasmic reticulum-Golgi intermediate complex. Finally, new viral particles are incorporated into vesicles and secreted by the host cells (*V'kovski et al., 2021*).

Viral infection triggers a variety of pathways in the host cells that ultimately lead to the hijacking of the cellular machineries escape from immune surveillance. SARS-CoV-2 infection elicits a peculiar gene expression response that first involves activation of interferon pathway (*Blanco-Melo et al., 2020*; *Vanderheiden et al., 2020*; *Wyler et al., 2021*), and then the NFkB pathway (*Kircheis et al., 2020*; *Hariharan et al., 2021*; *Wyler et al., 2021*), as well as expression of specific cytokines such as IL6 and IL8 (*Wang et al., 2007*; *Blanco-Melo et al., 2020*; *Coperchini et al., 2020*; *Park and Lee, 2020*).

Recent and intense research on SARS-CoV-2 characterized the role of viral proteins during viral replication and showed that these functions are often conserved across coronaviruses (*V'kovski et al., 2021*). Less is known about the roles of the individual proteins in modulating host cell pathways (*Gordon et al., 2020a*; *Gordon et al., 2020b*). For instance, recent studies have proposed that Nsp16 is a splicing modulator (*Banerjee et al., 2020*) and that Nsp1 and Nsp14 are translational repressors (*Schubert et al., 2020*; *Hsu, 2021*).

Nsp14 is a 60 kDa protein conserved among coronaviruses that is involved both in viral replication and in immune surveillance escape (*Ogando et al., 2020*). The N-terminal region of Nsp14 contains an exonuclease (ExoN) domain that excises mismatched nucleotides to ensure accurate replication of the viral genome (*Ogando et al., 2020*). As a result of this proofreading mechanism, coronaviruses have a lower mutation rate than other RNA viruses (error rate of $10^6$–$10^7$ vs. $10^3$–$10^5$) (*Sanjuán et al., 2010*; *Robson et al., 2020*). Loss of function of the ExoN domain results in increased sensitivity to the RNA mutagen 5-fluorouracil (*Eckerle et al., 2010*) and attenuated virulence (*Graham et al., 2012*). Furthermore, the interaction with Nsp10 augments Nsp14 ExoN activity up to 35-fold, and inhibition of the interaction between Nsp10 and Nsp14 leads to reduced replication fidelity (*Ma et al., 2015*; *Smith et al., 2015*). Nsp14 is also involved in assembly the cap at the 5' end of the viral RNA genome, which is crucial for evading immune surveillance. The C-terminal region of Nsp14 functions as an *S*-adenosyl methionine-dependent guanine-N7 methyl transferase that is independent of the ExoN activity (*Chen et al., 2009*). Both enzymatic domains are essential for successful viral replication, making Nsp14 an appealing drug target (*Otava et al., 2021*; *Saramago et al., 2021*).

Nsp14 is part of the replication complex, therefore it interacts with other SARS-CoV-2 proteins. As for other coronaviruses, replication of SARS-CoV-2 genome takes place in replication organelles that provide a protective environment for the newly synthesized viral genome (*V'kovski et al., 2021*). Notably, these organelles are formed in the cytoplasm and present convoluted double layered membranes that likely exchange material with the cytoplasm through pores (*Wolff et al., 2020*).

Furthermore, Nsp14 might mediate immune surveillance escape by activating the interferon pathway and activating the pro-inflammatory response through NFkB transcriptional activity and *CXCL8* expression (*Yuen et al., 2020*; *Hsu, 2021*; *Li et al., 2021*). However, the biological mechanism behind these events has not fully characterized. Furthermore, a global interactome study showed that Nsp14 interacts with the cellular enzyme inosine-monophosphate dehydrogenase 2 (IMPDH2) and that this interaction is conserved also in SARS-CoV-1 and MERS-CoV viruses (*Gordon et al., 2020a*; *Gordon et al., 2020b*). IMPDH2 catalyzes the conversion of inosine-5'-monophosphate (IMP) to xanthine-5'-monophosphate (XMP) (*Hedstrom, 2009*), which is the rate-limiting step of de novo guanine nucleotides biosynthesis. Guanosine-5'-triphosphate (GTP) is necessary for DNA replication and transcription and is used as energy source for translation and as mediator of signal transduction (*Hesketh and Oliver, 2019*). How the physical interaction between Nsp14 and IMPDH2 impacts the host pathways is not completely understood (*Li et al., 2021*).

Interestingly, a recent study has shown that expression of Nsp14 results in global translation inhibition, but it is not clear if this is a direct effect or a downstream consequence of the changes that the expression of this protein provokes to the cellular environment (*Hsu, 2021*). Indeed, no direct interaction between Nsp14 and ribosomes or a known translational modulator has been reported, suggesting that the potential translational inhibition might be a downstream effect.

Here, we undertook transcriptome analyses in cells that express each SARS-CoV-2 protein individually. Expression of Nsp14 altered the expression of about 4000 genes, mostly involved in splicing, RNA metabolism, and cell-cycle control. Importantly, the effect of Nsp14 on cellular gene expression resembled the transcriptional changes that occur upon SARS-CoV-2 infection and included the activation of the NFkB pathway and the expression of *CXCL8* (encoding IL8), a marker of acute severe respiratory distress syndrome in COVID-19 patients (*Adcock et al., 2015*; *Blanco-Melo et al., 2020*; *Kircheis et al., 2020*). Intriguingly, we also detected an increase in circRNAs expression upon Nsp14 expression; recent studies indicate that circRNAs can act as modulators of the innate immune response during viral infections (*Li, 2017*; *Liu, 2019*; *Chen et al., 2020*; *Yan and Chen, 2020*). Moreover, we showed that the cellular enzyme IMPDH2 mediates the gene expression response induced by Nsp14. We found that *IMPDH2* mRNA is downregulated upon Nsp14 expression and that the cellular GTP concentration strongly increases, indicating that Nsp14 might activate IMPDH2 enzymatic activity. In accordance with our hypothesis, we showed that treatment with IMPDH2 inhibitors (mycophenolic acid [MPA] and mizoribine [MZR], *Lee et al., 2020a*), partially rescued the changes in gene expression induced by Nsp14.

## Results

### Expression of individual SARS-CoV-2 proteins specifically remodels the transcriptome

Infection of cells with SARS-CoV-2 induces strong and specific changes in the transcriptome of host cells and tissues (*Blanco-Melo et al., 2020*; *Wyler et al., 2021*). It is assumed that this response results from the hijacking of the cellular systems by the virus as well as from the defense by the host. To identify unknown functions of the individual SARS-CoV-2 proteins and to determine how much each protein contributes to the takeover of cellular systems, we determined how the transcriptome changed when we individually express each viral protein in a human cell line. Specifically, we expressed individual SARS-CoV-2 proteins (*Gordon et al., 2020b*) in HEK293T cells, and after 48 hr we purified RNA, generated and sequenced 3' RNA sequencing (RNA-seq) libraries, and identified differentially expressed genes (DEGs; *Figure 1A*). Extensive cell death or changes in morphology did not occur upon expression of individual proteins, as assessed visually.

Interestingly, expression of most proteins resulted in modest or no changes in the transcriptome of the HEK293T cells. Indeed, we detected less than 300 DEGs upon expression of 17 of the 25 tested proteins (*Supplementary file 1a* and *Figure 1B*). Expression of seven viral proteins, M, Nsp9, E, ORF9b, ORF3a, Nsp13, and Nsp1, modestly altered the transcriptome (between 300 and 700 DEGs). Interestingly, these DEGs tended to be upregulated rather than downregulated (*Figure 1B*). Striking, Nsp14 altered the expression of more than 4000 RNAs (1862 upregulated and 2161 downregulated; *Figure 1B*). The profound impact of Nsp14 expression on the transcriptome of HEK293T

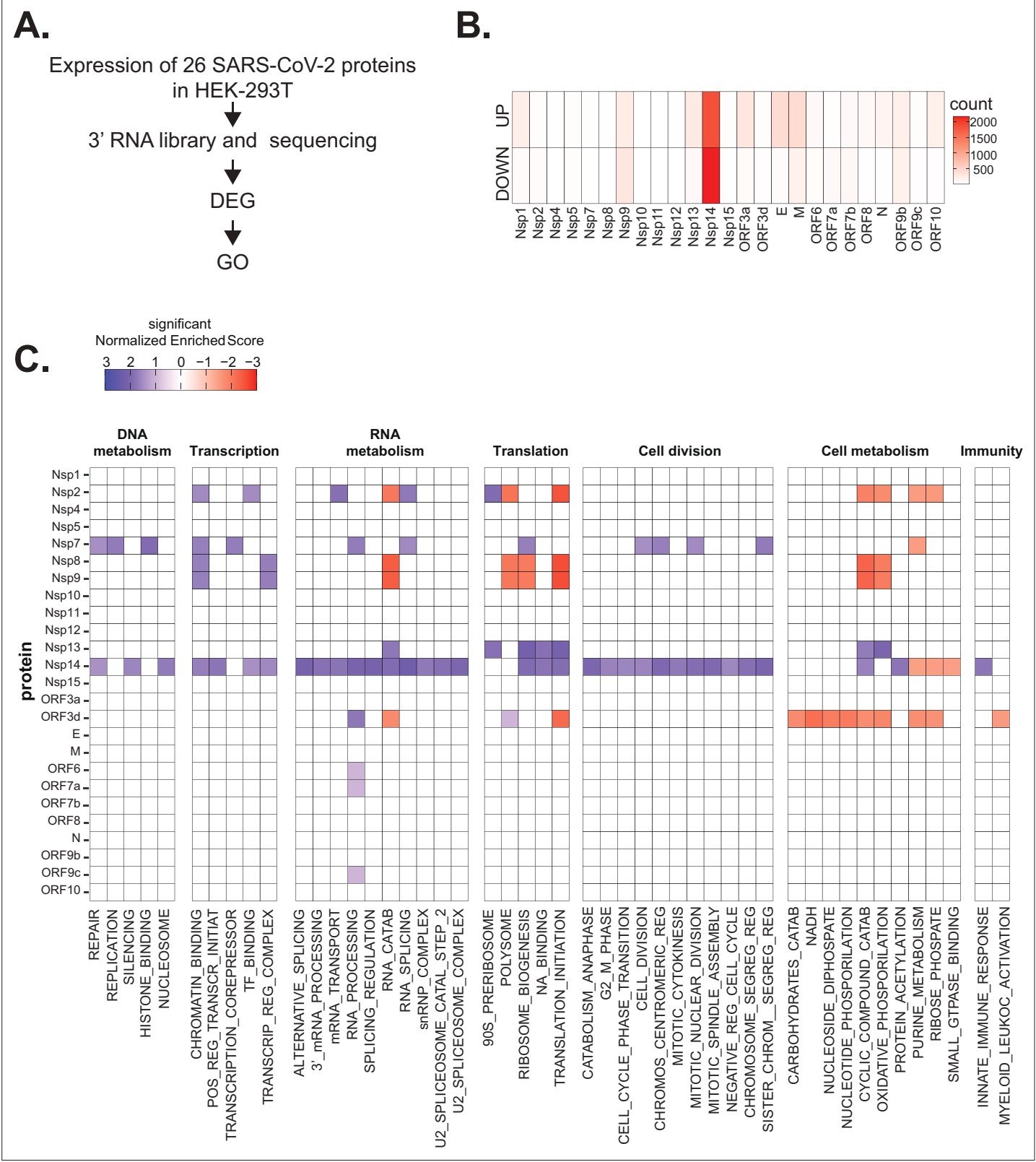

**Figure 1.** Severe acute respiratory syndrome coronavirus 2 (SARS-CoV-2) proteins alter gene expression distinctively. (**A**) Scheme of the experimental approach. DEGs stands for differentially expressed genes and GO for Gene Ontology. (**B**) Heatmap showing the number of DEGs detected in 3' RNA sequencing for each expressed SARS-CoV-2 protein. 'Up' stands for upregulated genes, 'Down' for downregulated genes. Lfc < 0.5, corrected p-value < 0.05. (**C**) Heatmap showing the GO analysis (colors represent the significant normalized enriched score).

cells suggests that this protein has roles beyond its known functions in viral genome proofreading and immune system escape.

To identify cellular pathways affected by the expression of the individual SARS-CoV-2 proteins, we performed Gene Ontology (GO) analysis of the DEGs upon expression of the different viral proteins (*Supplementary file 1b* and *Figure 1C*). Expression of the viral proteins impacted mRNAs encoding proteins related to different aspects of gene expression (regulation of transcription, translation, and RNA metabolism), cell metabolism, cell division, and innate immunity. For example, expression of Nsp7, an accessory protein of the RNA-dependent RNA polymerase (*Kirchdoerfer and Ward, 2019*; *Hillen, 2020*), deregulated expression of genes encoding proteins involved in DNA and RNA metabolism (*Figure 1C*). Expression of Nsp9 and Nsp13, known to bind RNA and to regulate RNA metabolism, respectively (*Egloff et al., 2004*; *Shu et al., 2020*), resulted in the mis-regulation of mRNAs encoding proteins involved in RNA metabolism and translation (*Figure 1C*). Furthermore, expression of the protein ORF3d, known to bind STOML2 mitochondrial protein, may be a unique antigenic target upon SARS-CoV-2 infection (*Gordon et al., 2020b*; *Hachim et al., 2020*; *Nelson et al., 2020*; *Jungreis et al., 2021*), resulted in alteration of genes involved in cell metabolism and immunity (*Figure 1C*). In general, expression of the viral proteins mainly impacted pathways related to RNA metabolism, translation, and cell metabolism that were previously reported to be affected upon viral infection (*Blanco-Melo et al., 2020*; *Gardinassi et al., 2020*; *Wyler et al., 2021*). Expression of Nsp14 affected the expression of genes involved in RNA splicing, metabolism, and processing, translation, cell-cycle control, and the cytoskeleton. Nsp14 also impacted the expression of genes involved in general metabolism, especially on genes implicated in nucleotide metabolism (*Figure 1C*).

## Nsp14 expression induces transcriptional changes that resemble SARS-CoV-2 infection

We decided to focus on Nsp14, as expression of this protein resulted in significantly more DEGs than any other tested protein. To obtain a more comprehensive view of the transcriptome changes provoked by Nsp14, we complemented the 3' RNA-seq data with total RNA-seq data generated from cells transfected with Nsp14. The new dataset strongly resembled the one obtained by 3' RNA-seq (*Figure 2A*, *Supplementary file 1c*, and *Figure 2—figure supplement 1A*) but also contained information about the full-length RNAs, non-poly-adenylated RNAs, and pre-mRNAs.

To gain further insights into the mechanism and pathways altered by Nsp14, we performed gene set enrichment analysis (GSEA) of the total RNA-seq data from Nsp14-expressing cells. Strikingly, we found that for genes upregulated upon expression of Nsp14, three of the four most highly enriched GSEA datasets were those of cells infected with SARS-CoV-2 (*Blanco-Melo et al., 2020*; *Figure 2B and C*, and *Figure 2—figure supplement 1B*). Moreover, we observed a smaller but significant enrichment of genes upregulated upon infection with MERS, a related virus (*Blanco-Melo et al., 2020*). Further, the mRNAs downregulated upon expression of Nsp14 tended to be strongly enriched for those downregulated following SARS-CoV-2 viral infection (*Figure 2B*). We did not detect enrichment when our dataset were compared to data from cells infected with influenza A virus (*Blanco-Melo et al., 2020*; *Figure 2B*). These results demonstrate that expression of a single viral protein can recapitulate a considerable portion of response of a host cell to SARS-CoV-2 infection and highlights the potential importance of Nsp14 in hijacking host gene expression.

## Nsp14 alters gene expression mostly at the transcriptional level

To evaluate whether the transcriptome changes were due to alterations at the transcriptional or posttranscriptional level or a combination of both, we utilized intronic signals from the total RNA-seq experiment as a proxy for transcriptional activity (*Lee et al., 2020b*). For each DEG in the 3' RNA-seq dataset, we determined whether the intronic signal in the total RNA-seq dataset was altered in the same direction. We observed a strong correlation ($R = 0.73$ and p-value = $2.2e^{-16}$) between these two measures (*Figure 2D*), indicating that a large part of the response is transcriptional. For example, of the 1862 genes that were upregulated upon Nsp14 expression, 1006 also had higher intronic signal (at least 20% signal increase), whereas only 57 of the showed decreased intronic signal. Similarly, of the 2161 genes downregulated upon Nsp14 expression, 1242 displayed lower intronic signal (at least 20% signal decrease), whereas only 109 showed higher intronic signal (*Supplementary file 1d*). We then performed a similar analysis using the exonic signal of DEGs in the same total RNA-seq dataset.

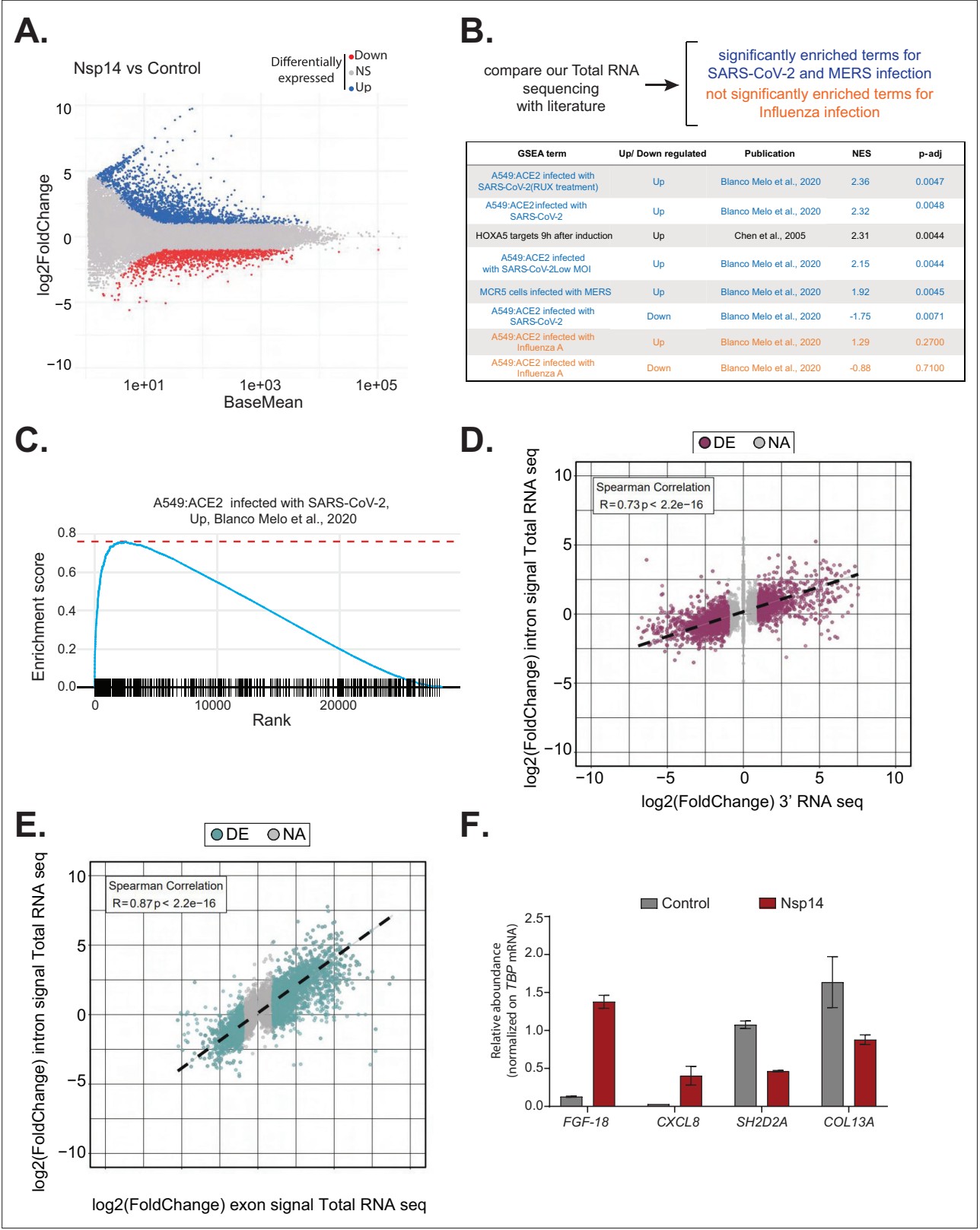

**Figure 2.** Expression of Nsp14 induces transcriptional changes like severe acute respiratory syndrome coronavirus 2 (SARS-CoV-2) infection. (**A**) MA plot showing the fold change of expression in samples expressing Nsp14 compared to control detected in the total RNA sequencing (RNA-seq). In red significantly downregulated genes, in blue upregulated genes, and in gray non significantly deregulated genes. (**B**) Scheme representing the approach to determine the overlap with our total RNA-seq data and already published dataset (top). Table reporting the gene set enrichment analysis (GSEA)

*Figure 2 continued on next page*

*Figure 2 continued*

terms, up- or downregulation, publication, the normalized enriched score (NES), and adjusted p-value (p-adj) when comparing our total RNA-seq data with previously published datasets. Significant terms related to SARS-CoV-2 and MERS infection are indicated in blue, non-significant terms related to influenza A infection are indicated in orange (bottom). (**C**) Example of GSEA. (**D**) Nsp14 expression vs. control fold change of intronic signal from total RNA-seq vs. 3' RNA-seq signal in logarithmic scale for each detected gene. Colored dots represent significantly changing genes (fold change = 2, adjusted p-value < 0.05, N = 3). (**E**) Nsp14 expression vs. control fold change of intronic signal from total RNA-seq vs. exonic signal from total RNA-seq in logarithmic scale for each detected gene. Colored dots represent significantly changing genes (fold change = 2, adjusted p-value < 0.05, N = 3). (**F**) RT-qPCR showing the abundance of *FGF-18, CXCL8, SH2D2A,* and *COL13A* in the chromatin-bound RNA fraction in cells transfected with an empty plasmid (control) or with Nsp14 (Nsp14). Data represented as mean ± SEM, N = 3.

The online version of this article includes the following figure supplement(s) for figure 2:

**Figure supplement 1.** Expression of Nsp14 induces transcriptional changes as SARS-CoV2 infection.

**Figure supplement 2.** Controls for subcellular fractionations and nascent RNA extractions.

We obtained an even stronger correlation between the changes in exonic signal and total RNA signal for each gene ($R = 0.87$ and p-value = $2.2e^{-16}$, *Figure 2E*). In sum, we observed that more than 50% of the changes in the transcriptome upon Nsp14 expression are transcriptional. Notably, this might be an underestimation, as changes in intronic signal could be lower than the steady-state RNA levels or some introns could be very efficiently and quickly removed, so that splicing intermediates are not detected. To further confirm the transcriptional effect, we determined the levels of a subset of DEGs from chromatin-bound nascent RNA from cells transfected with a control or a Nsp14-expressing plasmid. As expected, a pre-mRNA (*pre-TBP*), was enriched in the chromatin-bound fraction, *U6* was abundant in the nuclear compartment (nucleoplasm and chromatin bound), whereas *18S rRNA* and a circRNA (*circVKR1*) were more abundant in the cytoplasm (*Figure 2—figure supplement 2A*). Then, we checked the expression of some genes that were up or down regulated upon Nsp14 expression in our dataset in the chromatin-bound fraction. Genes upregulated upon Nsp14 expression (*FGF-18* and *CXCL8*) were also upregulated in the chromatin-bound fraction, while the downregulated ones (*SH2D2A* and *COL13A*) were downregulated indicating that at least those mRNAs are regulated at the transcriptional level by Nsp14. These results verify the genomic observations and strongly suggest that the gene expression changes observed upon Nsp14 are mainly transcriptional (*Figure 2F*). Interestingly, we also observed that there are more than 1000 genes that display higher intronic signals with no changes in gene expression, indicating that, in addition to the transcriptional effects, Nsp14 might also affect splicing (*Supplementary file 1e*).

## Nsp14 expression provokes changes in alternative splicing and circRNAs production

Genes upregulated upon Nsp14 expression are enriched in genes encoding proteins with GO terms related to RNA metabolism and, more specifically, splicing (*Figure 1C*). Moreover, for more than 1000 genes, there were significant increases in intron signal upon expression of Nsp14 without changes in mRNA levels, suggesting that Nsp14 alters the splicing of these pre-mRNAs (*Supplementary file 1e*). Indeed, we found that expression of Nsp14 strongly altered the inclusion patterns of almost 2300 exons, with more than 2000 exons displaying lower inclusion in the mature mRNA and 238 showing higher levels of inclusion when we expressed Nsp14 (*Figure 3A*, *Supplementary file 1e*, and *Figure 3—figure supplement 1A*). Furthermore, we also identified genes which used alternative acceptor or donor splice sites upon expression of Nsp14 (*Figure 3A* and *Supplementary file 1e*). Moreover, we observed an increase in the retention of more than 2000 introns following Nsp14 expression (*Figure 3A and C*). Although the effects on exon inclusion and use of alternative splice sites clearly indicate that Nsp14 influences splicing, the increase in intron retention could be secondary to changes in transcription. However, as most of the introns with increased retention are within genes that were not differentially expressed (*Figure 3B* and *Figure 3—figure supplement 1B*), we reasoned that expression of Nsp14 leads to changes in splicing in this subset of genes as well. Moreover, the effect of Nsp14 on alternative splicing appears to be specific to particular introns and exons in each gene, as the majority (~62%) of identified genes had a single spicing event altered (*Figure 3D* and *Supplementary file 1e*). Notably, most genes with altered splicing do not show expression changes upon Nsp14 expression, further indicating that the detected alternative splicing events are independent to transcriptional changes (*Figure 3—figure supplement 1B*).

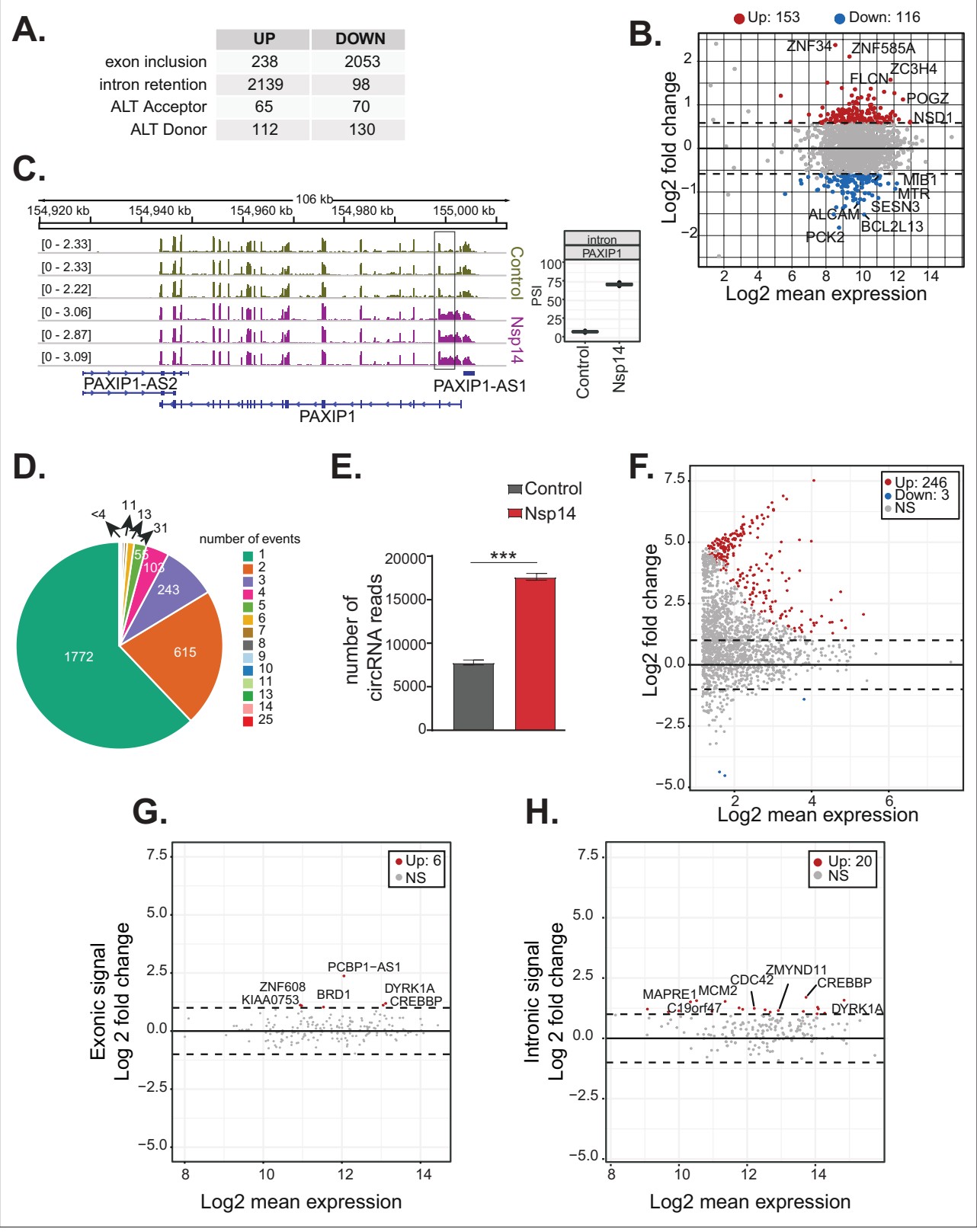

**Figure 3.** Nsp14 expression alters the splicing of a subgroup of genes and increases circRNAs expression. (**A**) Table summarizing splicing analysis comparison between Nsp14 expression and control. Thresholds used: ΔPSI (percentage of inclusion) > 15% and a non-overlapping distribution with minimum of 5% difference (N = 3). (**B**) Fold change vs. expression in logarithmic scale for the genes with upregulated intron retention. In red genes with increased expression and in blue the ones with downregulated expression (fold change = 2, adjusted p-value < 0.05, N = 3). (**C**) Representative

*Figure 3 continued on next page*

*Figure 3 continued*

IGV alignment tracks of on gene (PAXIP1) with intronic events differentially changing between conditions (control and Nsp14 expression). The box marks the changing event. On the right, quantification of PSI. (**D**) Pie chart representing number of alternative splicing events deregulated upon Nsp14 expression by gene; 1772 genes have only one alternative splicing event changing between conditions, 615 has two events and 243 genes have three alternative splicing events changing. (**E**) Number of circRNAs reads detected in the total RNA sequencing (RNA-seq) experiment. Data represented as mean ± SEM, N = 3, t-test, ***p-value < 0.0005. (**F**) Fold change vs. expression in logarithmic scale for circRNAs in Nsp14 expression vs. control. In red upregulated genes and in blue downregulated genes (fold change = 2, adjusted p-value < 0.05, N = 3). (**G**) Plot of fold change vs. expression in logarithmic scale for exonic signal detected in the total RNA-seq dataset in Nsp14 vs. control for genes with upregulated circRNA expression. In red genes with increased expression and (fold change = 2, adjusted p-value < 0.05, N = 3). (**H**) Plot of fold change vs. expression in logarithmic scale for intronic signal detected in the total RNA-seq dataset in Nsp14 vs. control for genes with upregulated circRNA expression. In red genes with increased expression and in gray non-significant ones (fold change = 2, adjusted p-value < 0.05, N = 3).

The online version of this article includes the following figure supplement(s) for figure 3:

**Figure supplement 1.** Examples and features of genes showing altered splicing upon Nsp14 expression.

To further characterize the effects of Nsp14 expression on alternative splicing, we looked for genomic features associated with the altered patterns of splicing in the presence of Nsp14. Interestingly, we found that the affected introns tended to be at the 5′ end of the transcript and were embedded in genomic regions with higher-than-average GC content (*Figure 3—figure supplement 1C* and *Supplementary file 1f*). The GC content may be a confounding factor of the location of these introns, as the GC content is higher at the 5′ end of genes and around the transcription start sites (*Zhang, 2004*). Therefore, we concluded that Nsp14 expression has a strong and specific effect on splicing efficiency for a subset of genes.

Furthermore, most of the affected exons are shorter and have higher GC content around their splice sites than a randomized subset of exons not affected by Nsp14 (*Figure 3—figure supplement 1D* and *Supplementary file 1f*). The higher GC content suggests that stable RNA structures form around the splice sites in these exon, which correlates with alternative splicing propensity (*Zhang et al., 2011*; *Lin et al., 2016*).

Previous research showed that SARS-CoV-2 infection disrupts splicing, mostly by inducing intron retention (*Banerjee et al., 2020*). To check whether the effects on splicing induced by Nsp14 were comparable to the ones that happen during SARS-CoV-2 infection, we re-analyzed a published dataset of total RNA-seq from HEK293T cells expressing human ACE2 and infected with SARS-CoV-2 (*Sun et al., 2021*). We found that there is a significant overlap (p-value < 0.01, see Materials and methods) between the alternative splicing events during the infection and in our model. Specifically, 10% of the alternative splicing events changing with SARS-CoV-2 infection (*Supplementary file 1g*) also change (and in the same direction) when we express Nsp14 (*Supplementary file 1e*). In sum, we showed that expression of Nsp14 partially recapitulates the gene expression changes, as well as some of the alternative splicing events that occur upon SARS-CoV-2 infection.

As circRNAs are generated by back-splicing, a process that competes with linear pre-mRNA splicing (*Ashwal-Fluss et al., 2014*), we checked whether Nsp14 also alters circRNAs expression. Using the total RNA-seq data, we found that there is a strong increase (more than twofold) in the total number of circRNA reads upon Nsp14 expression (*Figure 3E*). Most of the 246 circRNAs that were differentially expressed upon expression of Nsp14 were upregulated (put the number of upregulated; *Figure 3F* and *Supplementary file 1h*). These deregulated circRNAs were contained within 190 loci (some loci host multiple circRNAs). Interestingly, the levels of most of the mRNAs that host these circRNAs were unchanged (*Figure 3G* and *Supplementary file 1d*). Indeed, Nsp14 did not increase the transcription of loci hosting the upregulated circRNAs (we observed increased levels of intronic sequences in only 20 out of the 190 loci with upregulated circRNAs; *Figure 3H* and *Supplementary file 1d*). Together, these results strongly suggest that the increased circRNA levels are caused by increased biosynthesis or stability of the circRNAs. In sum, our data shows that expression of Nsp14 influences alternative splicing and back-splicing on the host cells.

## The effect of Nsp14 on gene expression is independent of the ExoN activity but requires the N7-guanine-methyltransferase domain

Nsp14 has two separated enzymatic activities: (1) works as a proofreading ExoN and (2) it is a N7-guanine-methyltransferase required for the modification of the viral RNA cap. To determine whether the ExoN

activity is required for the strong effects of Nsp14 on gene expression, we performed two different experiments. First, we tested whether co-expression of Nsp10, which dramatically increases the ExoN activity of Nsp14 (*Bouvet et al., 2012*; *Bouvet et al., 2014*; *Ma et al., 2015*), resulted in an enhanced effect of Nsp14. Briefly, we transfected HEK293T cells with plasmids for expressing Nsp14 and/or Nsp10. As a control, we used cells transfected with an empty vector. Co-expression of Nsp14 and Nsp10 did not impact levels of Nsp14 protein (*Figure 4A*). We then prepared RNA from these samples and quantified the expression of some mRNAs that were altered upon expression of Nsp14 alone. For instance, the levels of *FGF-18* increased upon expression of Nsp14 alone, whereas the levels of *SH2D2A* were significantly downregulated (*Figure 4—figure supplement 1A*). Interestingly, expression of Nsp10 alone or in combination with Nsp14 did not further alter the levels of any of the mRNAs tested (*Figure 4—figure supplement 1A*). To determine whether the effect is global, we prepared and sequenced total RNA from these samples. Principal component analysis (PCA) showed that data from samples expressing Nsp14 alone separated from the data from the sample transfected with the empty plasmid, whereas that the ones from the samples expressing only Nsp10 almost overlapped with the control. Co-expression of Nsp14 and Nsp10 resulted in transcriptional changes like the ones induced by expression of only Nsp14 (*Figure 4—figure supplement 1B*). Unlike expression of Nsp14, which provoked a strong remodeling of the mRNA and circRNA populations (*Figures 1–3*), expression of Nsp10 did not considerably alter the gene expression profile (only 88 DEGs, *Figure 4B and C*, and *Figure 4—figure supplement 1C*). As also suggested by the RT-qPCR experiments (*Figure 4—figure supplement 1A*), the gene expression profiles of cells transfected with Nsp14 alone or in combination with Nsp10 are strikingly similar. Indeed, we observed an almost complete overlap between the DEGs resulting from Nsp14 expression and from co-expression of Nsp14 and Nsp10, and there was a very strong correlation between the fold change in DEGs upon Nsp14 expression with and without Nsp10 co-expression (*Figure 4B and C*). Moreover, co-expression of Nsp10 did not result in additional differentially expressed mRNAs (*Figure 4B and C*) and did not alter of the number of circRNAs reads (*Figure 4D*) compared to expression of Nsp14 alone. These data strongly suggests that the Nsp14 ExoN activity is not responsible for the dramatic effects of Nsp14 expression on the transcriptome.

To definitely test whether the ExoN activity is related to the remodeling of gene expression upon Nsp14 expression, we determined the effect of Nsp14 ExoN mutants on gene expression. Briefly, we generated plasmids for expression of Nsp14 with mutations in the ExoN domain: the double mutant D90A/G92A and the single mutant D273A (*Figure 4E*, schematic *Figure 4—figure supplement 2A*). These mutations have been previously shown to completely inhibit the ExoN activity of Nsp14 (*Ma et al., 2015*; *Ogando et al., 2020*). We generated and sequenced 3′ RNA-seq libraries from cells that expressed wild-type or ExoN-mutant Nsp14 proteins or eGFP, as a control. PCA indicated that the two mutants have overlapping transcriptomes that differed from the control (*Figure 4—figure supplement 2B*). The mutations in the ExoN domain did not alter the strong effects induced by Nsp14 expression (*Figure 4—figure supplement 2C*) and the fold changes in mRNA expression induced upon expression of the wild-type or mutated Nsp14 were also strikingly similar (*Figure 4E*). Moreover, we did not detect significant differences in the expression of the three tested circRNAs between samples that expressed the wild-type Nsp14 (Nsp14 WT) and the ExoN mutants (*Figure 4—figure supplement 2D*). Taken together, these data indicate that the ExoN activity of Nsp14 is not required for the observed transcriptome effects.

We also generated a N7-guanine-methyltransferase mutant (Nsp14 D331A), by altering a conserved site previously shown to be essential for the N7-guanine-methyltransferase activity (*Jin et al., 2013*) (schematic *Figure 4—figure supplement 2A*). Then, we checked by RT-qPCR how Nsp14 D331A affects the expression of some mRNA and circRNAs that are deregulated with Nsp14 WT. Interestingly, overexpression of Nsp14 D331A does not alter the expression of the tested targets (*Figure 4G and H*). The lack of effect is not due to low expression or stability of the Nsp14 D331A mutant, as it was expressed at similar levels than Nsp14 WT as assessed by Western blot (*Figure 4—figure supplement 2E*). In summary, our results indicate that the N7-guanine-methyltransferase domain of Nsp14 is required for driving the gene expression changes described in this study.

## Dissecting the transcriptome changes upon Nsp14 expression

To gain further insights into the gene expression program orchestrated by Nsp14, we performed a time course experiment. Briefly, we prepared and sequenced 3′ RNA-seq libraries using RNA isolated

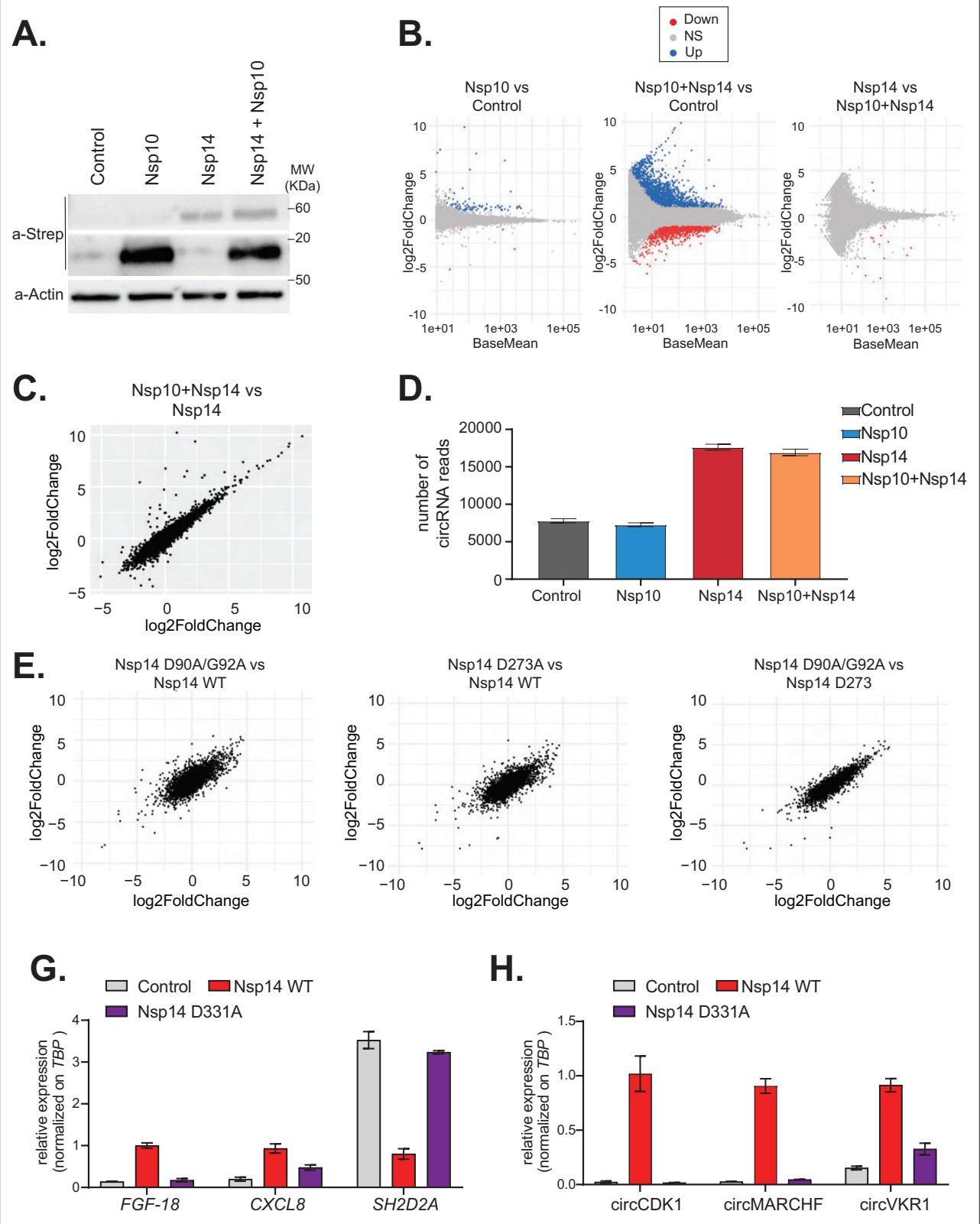

**Figure 4.** The N7-guanine-methyltransferase domain but not the exonuclease activity of Nsp14 is required for changing gene expression. (**A**) Western blot in cells transfected with an empty plasmid (control), Nsp10, Nsp14, or co-expressing Nsp10 and Nsp14 (Nsp10 + Nsp14). Nsp10 and Nsp14 were detected through the Strep tag. Actin was used as loading control. See *Figure 4—source data 1*. (**B**) MA plots showing the expression fold change in-between the indicated conditions in the total RNA sequencing (RNA-seq) dataset. Significantly upregulated genes in blue, downregulated in red, and

*Figure 4 continued on next page*

*Figure 4 continued*

not significantly deregulated in gray. (**C**) Plot showing the fold change of deregulated genes in samples co-expressing Nsp10 and Nsp14 vs. control (on the y-axis) and Nsp14 vs. control (on the x-axis). (**D**) Number of circRNAs reads detected in each indicated condition. Data represented as mean ± SEM, N = 3. (**E**) Plots showing the fold change of deregulated genes in the indicated condition vs. control. (**G**) RT-qPCR showing the expression of *FGF-18*, *CXCL8*, and *SH2D2A* upon the transfection with an empty plasmid (control), Nsp14 WT, or Nsp14 D331A. Data represented as mean ± SEM, N = 3. (**H**) RT-qPCR showing the expression of circCDK1, circMARCHF7, and circVKR1 upon the transfection with an empty plasmid (control), Nsp14 WT, or Nsp14 D331A. Data represented as mean ± SEM, N = 3.

The online version of this article includes the following source data and figure supplement(s) for figure 4:

**Source data 1.** Western blot data for *Figure 4A*.

**Figure supplement 1.** Co-expression of Nsp10 does not change the effects induced by Nsp14.

**Figure supplement 2.** The N7-guanine-methyltransferase activity but not the exonuclease one of Nsp14 is required for changing gene expression.

**Figure supplement 2—source data 1.** Western blot data for *Figure 4—figure supplement 2E*.

from cells transfected with the Nsp-14-expressing plasmid 6, 8, 12, 24, and 48 hr after transfection. As a control, we collected data from cells transfected with a control plasmid (eGFP) harvested at the same time points. Importantly, we observed detectable levels of Nsp14 mRNA and protein even early after transfection (*Figure 5—figure supplement 1A and B*). As expected, the biological triplicates mapped together in the PCA (*Figure 5A*). Data diverged across time and between control and Nsp14-expressing cells. The effect of Nsp14 expression dramatically increased over time with a gradual increase in the number of DEGs (*Figure 5B*, *Figure 5—figure supplement 1C*, and *Supplementary file 2a*), as well as an increase in the number of significantly enriched GO terms (*Figure 5C* and *Supplementary file 2b*). For example, expression of genes related to cell metabolism, RNA metabolism, and cell cycle were altered relative to the control only at 24 and 48 hr post transfection (*Figure 5C*).

We reasoned that the mRNAs that were differentially expressed at early time points could provide cues regarding the mechanism of action of Nsp14. We observed only 19 DEGs at the 8 hr time point and 200 DEGs at the 12 hr time point (*Figure 5B*). Interestingly, *CXCL8*, which encodes IL8, was the most highly induced mRNA at the 8 hr time point (fold change = 11.5, p-value = $1.2e^{-5}$, *Supplementary file 1h*). Furthermore, we found that *CXCL8* was induced by more than 12-fold consistently between 8 and 48 hr post transfection (*Figure 5D*). It was previously reported that *CXCL8* mRNA is significantly upregulated following infection of cells with SARS-CoV-2 and that the levels of IL8 are increased in COVID-19 patients (*Blanco-Melo et al., 2020*; *Coperchini et al., 2020*; *Del Valle et al., 2020*; *Gardinassi et al., 2020*; *Li et al., 2021*).

To investigate whether Nsp14 expression induces the transcription of *CXCL8*, we generated a Firefly luciferase reporter under the control of the CXCL8 promoter (–500 to +80 bp around the transcription start site, *Figure 5—figure supplement 1E*). Expression of Nsp14 led to an almost fourfold increase in the signal from the luciferase reporter (*Figure 5E*), indicating that Nsp14 expression activates *CXCL8* transcription.

*CXCL8* expression is controlled by several transcription factors (TFs), including NFkB, a master regulator of inflammation (*Kunsch and Rosen, 1993*). Recent reports have shown that NFkB is activated during infections with SARS-CoV and SARS-CoV-2, although the mechanism and the importance of this activation are not well established (*Liao, 2005*; *Kircheis et al., 2020*; *Park and Lee, 2020*; *Hariharan et al., 2021*; *Hsu, 2021*; *Li et al., 2021*). Strikingly, some of the top upregulated RNAs at 8 hr post transfection of Nsp14 were *NFKBIA*, *JUN*, and *ATF3* (*Figure 5—figure supplement 1D*), which encode proteins involved in the activation and modulation of the NFkB pathway (*Stein et al., 1993*; *Baldwin, 1996*; *Pahl, 1999*).

To determine if NFkB activation contributes to the early response of cells following Nsp14 expression, we performed an unbiased analysis for TF binding site enrichment on the promoter regions of the genes with strong upregulation at the 8 hr time point. We found a significant enrichment in those promoters for binding sites recognized by components of the NFkB transcription complex and the NFkB regulatory network (*Figure 5F*, *Supplementary file 2c, and d*, *Stein et al., 1993*; *Baldwin, 1996*; *Pahl, 1999*). We next tested whether expression of Nsp14 activates a luciferase reporter containing a minimal promoter and NFkB sites (*Wilson et al., 2013*). Indeed, we observed that expression of

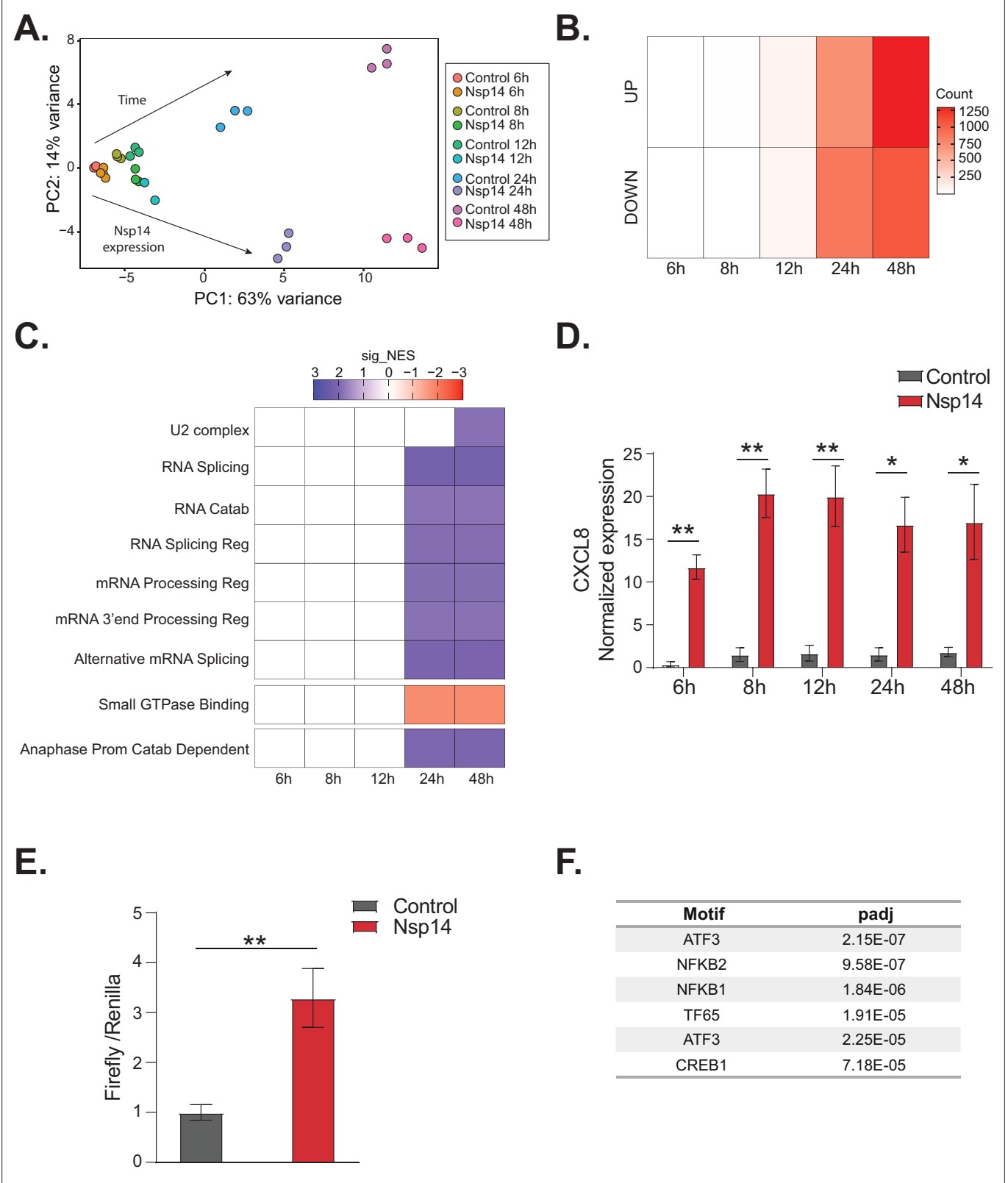

**Figure 5.** NFkB pathway is activated soon after Nsp14 transfection. (**A**) Principal component analysis of the 3' RNA sequencing (RNA-seq) of the time course experiment. Arrows indicate how samples separate according to the time point or condition (Nsp14 expression or control). (**B**) Heatmap showing increasing number of up- and downregulated genes at different time points after Nsp14 expression. (**C**) Heatmap representing the Gene Ontology analysis result at the indicated time points. (**D**) Expression of *CXCL8* across the indicated time points in the 3' RNA-seq data. Data represented as mean

*Figure 5 continued on next page*

*Figure 5 continued*

± SEM, N = 3, t-test, **p-value < 0.005, *p-value < 0.05. (E) Luciferase assay showing that CXCL8 is transcriptionally activated after Nsp14 expression. Firefly expression is controlled by CXCL8 promoter, whereas Renilla is under the control of a constitutive promoter. Data represented as mean ± SEM, N = 6, t-test, **p-value < 0.005. (F) Table showing the enrichment for specific transcription factor (TF) binding sites in the promoter (100 bp upstream the transcription starting site of the upregulated genes) (lfc >0.8, adjusted p-value < 0.05) at 8 hr after Nsp14 transfection.

The online version of this article includes the following source data and figure supplement(s) for figure 5:

**Figure supplement 1.** NFkB pathway is one of the first pathways activated after Nsp14 transfection.

**Figure supplement 1—source data 1.** Western blot data for *Figure 5—figure supplement 1B*.

Nsp14 lead to an approximately 1.6-fold increase in the luciferase signal (*Figure 5—figure supplement 1F*), demonstrating that this viral protein activates the NFkB pathway.

Then, for the mRNAs strongly upregulated early after Nsp14 transfection, we performed an analysis of their extended promoter regions (1000 bp upstream of the transcription start site). We found that these regions were enriched for binding sites for factors involved in the interferon response such as EGR2, STAT2, and SP3 (*Supplementary file 2e*). Previous work showed that SARS-CoV-2 infection modulates both the interferon and the NFkB pathways (*Blanco-Melo et al., 2020*; *Vanderheiden et al., 2020*; *Wyler et al., 2021*) and that Nsp14 affects these pathways (*Hsu, 2021*; *Li et al., 2021*). Our data indicates that the initial response to Nsp14 expression involves modulation of both the NFkB and the interferon pathways.

## IMPDH2 partially mediates the transcriptional changes induced by Nsp14 expression

Given the strong transcriptional changes we observed upon Nsp14 expression, it is possible that this viral protein acts directly as a TF. However, previous research showed that Nsp14 localizes in the cytoplasm when expressed in HEK293T cells (*Gordon et al., 2020a*). We then performed subcellular fractionation and chromatin purification, followed by Western blot at different time points after transfection, and we showed that Nsp14 localizes in the cytoplasm, and it does not associate with chromatin demonstrating that the transcriptional effect is indirect (*Figure 6—figure supplement 1A*).

Therefore, we explored the hypothesis that host cellular proteins mediate Nsp14-induced gene expression changes. Recent proteomics studies found that Nsp14 interacts with the host protein IMPDH2 (*Gordon et al., 2020a*; *Gordon et al., 2020b*), which catalyzes the rate-limiting step of guanine biosynthesis, but did not reveal whether this interaction results in any metabolic or cellular change (*Li et al., 2021*). One possibility is that Nsp14 activates IMPDH2 which then modulates gene expression directly or indirectly. In support of this possibility, expression of Nsp14 changed the levels of genes involved in purine metabolism (*Figure 1C*). Furthermore, we found *IMPDH2* mRNA expression significantly downregulated upon Nsp14 expression (*Figure 6A*). As GTP levels inversely regulate *IMPDH2* expression (*Glesne et al., 1991*), we hypothesized that Nsp14 promotes the activation of IMPDH2, leading to higher GTP levels that, in the end, negatively regulate *IMPDH2* mRNA. To test whether expression of Nsp14 alters IMPDH2 activity, we performed metabolomic analysis of lysates of cells transfected with a plasmid for expression of Nsp14 or a control plasmid. Consistent with modulation of IMPDH2 activity by Nsp14, we detected an approximately threefold increase in the levels of xanthine diphosphate and GTP (*Figure 6B* and *Supplementary file 3a*), the downstream products of XMP. Furthermore, we observed increased concentrations of other nucleotides that directly or indirectly derive from IMP (*Figure 6B* and *Supplementary file 3a*), probably due to activated compensatory pathways.

To determine if IMPDH2 activity mediates the Nsp14-driven changes in gene expression, we compared the gene expression changes upon Nsp14 expression in the presence of an IMPDH2 inhibitor. Briefly, we transfected cells with a Nsp14 or with a control plasmid (eGFP) and 8 hr later treated the cells with MPA, a non-isoform-selective pan-IMPDH inhibitor, or DMSO. We harvested the cells 40 hr after the treatment. We used a concentration of MPA of 0.5 µM to avoid cytotoxic effects (*Qasim et al., 2011*). Although we treated the cells with a low concentration of MPA, MPA treatment could reduce the concentration of cellular GTP, which is the primary energy source for mRNA translation. We confirmed that MPA treatment did not alter Nsp14 or IMPDH2 levels by Western blot (*Figure 6C*).

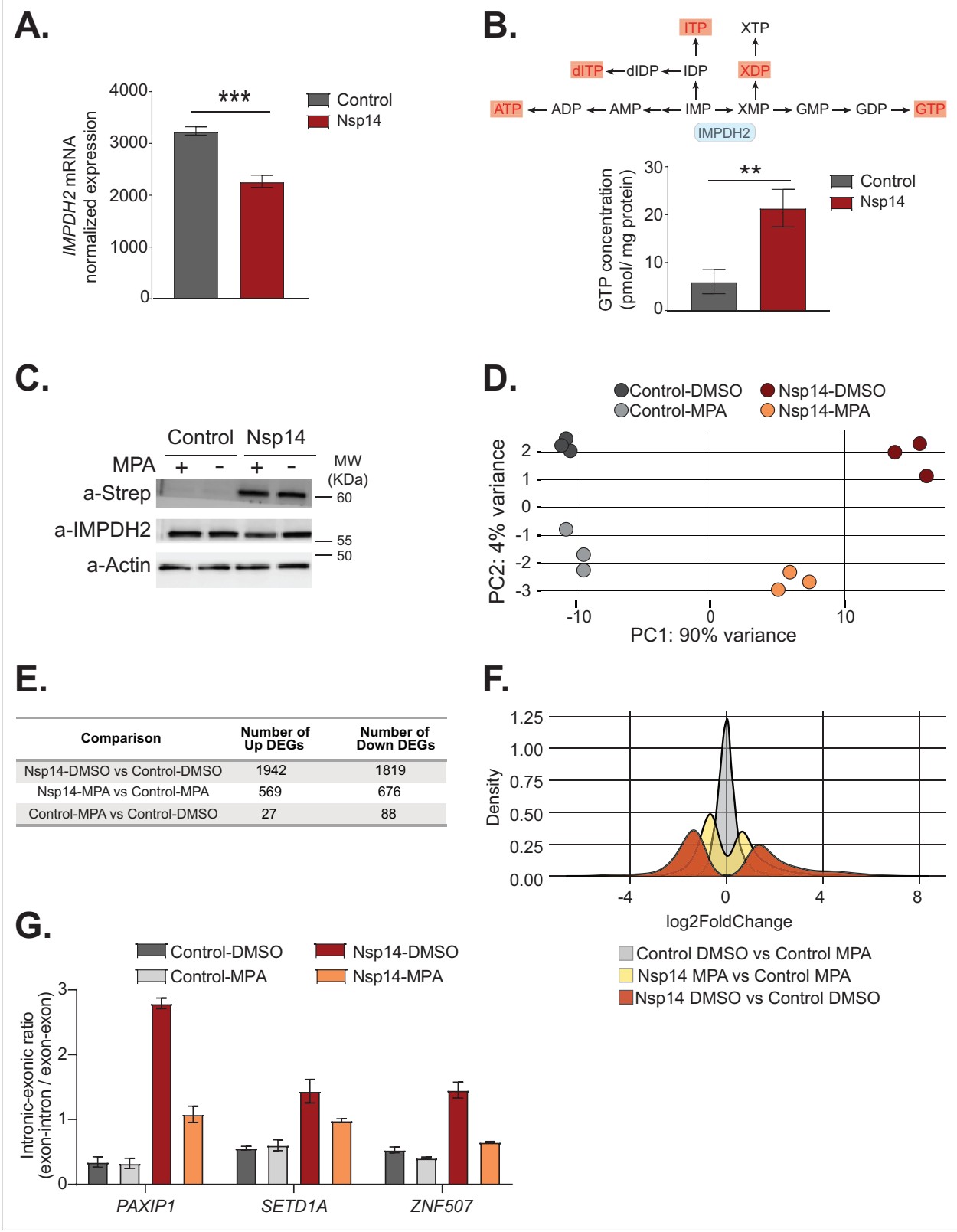

**Figure 6.** Pharmacological inhibition of inosine-monophosphate dehydrogenase 2 (IMPDH2) partially reverts the transcriptional changes induced by Nsp14. (**A**) Expression of *IMPDH2* mRNA is reduced upon Nsp14 expression. Data from the total RNA sequencing (RNA-seq) experiment. Data represented as mean ± SEM, N = 3, t-test, ***p-value < 0.0005. (**B**) In the upper panel, scheme reporting some of the tested metabolites deriving from inosine-5'-monophosphate (IMP) metabolism. IMPDH2 (highlighted in light blue) catalyzes the conversion of IMP to xanthine-5'-monophosphate

*Figure 6 continued on next page*

*Figure 6 continued*

(XMP), precursor of guanosine-5'-triphosphate (GTP). Significantly upregulated metabolites are highlighted in red. In the lower panel, GTP cellular concentration significantly increases in Nsp14-expressing cells. Data represented as mean ± SEM, N = 3, t-test, **p-value < 0.005. (**C**) Western blot showing that mycophenolic acid (MPA) treatment does not alter Nsp14 (detected through the Strep-tag) or IMPDH2 expression. Actin used as loading control. See *Figure 6—source data 1*. (**D**) Principal component analysis of the 3' RNA-seq library of the indicated samples. (**E**) Table reporting the number of upregulated and downregulated genes in the indicated comparisons. (**F**) Plot showing the distribution of fold changes of all genes detected in the 3' RNA-seq in the indicated conditions. (**G**) RT-qPCR showing the retention of the first intron for *PAXIP1*, *SETD1A*, and *ZNF507* in the indicated conditions. Data represented as mean ± SEM, N = 3.

The online version of this article includes the following source data and figure supplement(s) for figure 6:

**Source data 1.** Western blot data for *Figure 6C*.

**Figure supplement 1.** Nsp14 localizes in the cytoplasm and IMPDH2 mediates the effects induced by Nsp14.

**Figure supplement 1—source data 1.** Western blot data for *Figure 6—figure supplement 1A*.

**Figure supplement 2.** Inhibition of IMPDH2 partially reverts the changes induced by Nsp14.

**Figure supplement 2—source data 1.** Western blot data for *Figure 6—figure supplement 2D*.

We next performed 3' RNA-seq on the same samples. Interestingly, we observed that MPA treatment diminishes the effects induced by expression of Nsp14 (*Figure 6D*, *Figure 6—figure supplement 1B*, and *Figure 6—figure supplement 2A*). We detected only 115 DEGs in cells treated with MPA compared to control cells. In cells expressing Nsp14, we detected 3761 DEGs. By contrast, there were only 1245 DEGs in cells expressing Nsp14 that we treated with MPA compared to control cells, demonstrating that about 73% of the changes depends on the activity of IMPDH2 (*Figure 6E*). Approximately 1000 genes were deregulated in cells expressing Nsp14 with or without MPA (*Figure 6—figure supplement 2B*). In addition, the effect of those genes still differentially expressed was smaller in the presence of MPA. This can be seen by comparing the distribution of fold changes of the DEGs (*Figure 6F*). When we treated cells expressing Nsp14 with MPA, the distribution of fold change narrowed, indicating that MPA treatment mitigates the effects of Nsp14 even in those mRNAs that are still differentially expressed in the presence of the drug. Moreover, we observed a similar, milder, effect of Nsp14 expression when assaying the levels of a subset of circRNAs (*Figure 6—figure supplement 2C*). Furthermore, we corroborated these data using MZR, another IMPDH2 inhibitor. As for the MPA treatment, MZR treatment did no alter Nsp14 or IMPDH2 protein level (*Figure 6—figure supplement 2D*) and partially rescued the expression of the tested mRNAs and circRNAs deregulated by Nsp14 overexpression (*Figure 6—figure supplement 2E and F*). Together, the data presented here demonstrate that IMPDH2 is a key mediator of the effects of Nsp14 on the transcriptome of the hosting cell.

Finally, we investigated whether IMPDH2 also mediates the effects induced by Nsp14 on alternative splicing. To do so, we treated cells with MPA or MZR and we checked by RT-qPCR the abundance of some of transcripts that displayed intron retention in the total RNA-seq data (*Figure 3B* and *Supplementary file 1e*). As expected, we detected increased abundance of retained introns for *PAXIP1*, *SETD1A*, and *ZNF507* upon Nsp14 expression, and both MPA and MZR treatment partially rescued this effect, indicating that IMPDH2 mediates the observed splicing changes (*Figure 6G* and *Figure 6—figure supplement 2G*).

In summary, these results indicate that IMPDH2 mediates both the gene expression changes and the alternative splicing effects induced by Nsp14.

## Discussion

Viral infection leads to a complex set of events orchestrated by multiple viral proteins. In the last 2 years, new studies shed light in some unexpected functions of diverse SARS-CoV-2 proteins. For instance, protein interactome experiments defined the interaction map of SARS-CoV-2 proteins with host cellular proteins (*Gordon et al., 2020a*; *Gordon et al., 2020b*). Moreover, further studies showed that Nsp1 blocks cellular translation (*Schubert et al., 2020*), whereas RNA immunoprecipitation experiments indicated that Nsp8 and Nsp19 alter protein trafficking (*Banerjee et al., 2020*), and that Nsp16 alters splicing (*Banerjee et al., 2020*).

Here, we used an alternative approach; we performed an expression screen to determine how the individual expression of SARS-CoV-2 proteins influenced host cell gene expression. Nsp14 displayed the most striking effects, deregulating the expression of approximately 4000 mRNAs and altering splicing. While we acknowledge that the conditions under which we performed our experiments might lead to higher levels of Nsp14 than those observed during infection, we are convinced that our findings are highly relevant. Indeed, both the effects on gene expression and alternative splicing significantly overlap with the ones occurring during SARS-CoV-2 infection. Furthermore, during the infection Nsp14 localizes in the replication organelles that are formed in the cytoplasm (*V'kovski et al., 2021*), and in our model we also see that Nsp14 localizes in the cytoplasm. Overall, our data indicate that our model, despite some limitations, recapitulates some gene expression changes that occur during SARS-CoV-2 infection.

Regarding the other tested proteins, when expressed individually M, Nsp9, E, ORF9b, ORF3a, Nsp13, and Nsp1 also altered the host cells transcriptome in ways that will require further investigation. Although most of the tested proteins did not dramatically alter gene expression, this does not rule out their roles in remodeling of host cell gene expression. For instance, some may require other viral proteins as co-factors. In the future, it will be interesting to test how different combinations of viral proteins that are known to work in complex may affect host cell gene expression.

Our findings unrevealed an unexpected function of the SARS-CoV-2 Nsp14 in regulation of host cell transcription, splicing, and circRNAs expression. Notably, we also showed that IMPDH2 mediates these effects, as pharmacological inhibition of this cellular enzyme partially reverts them.

Our analysis of the transcriptomes of HEK293T cells as a function of time after Nsp14 expression revealed that the transcriptional response is temporally subdivided into two phases. The early response involves NFkB activation, whereas the later phase includes effects on cellular metabolism, RNA metabolism, and cell-cycle control. Notably, NFkB is one of the first pathways activated upon SARS-CoV-2 infection (*Li et al., 2021*; *Wyler et al., 2021*). Furthermore, early after transfection of Nsp14, we detected increased *CXCL8* expression, which encodes IL8. IL8 is elevated in plasma of COVID-19 patients (*Blanco-Melo et al., 2020*; *Hariharan et al., 2021*; *Li et al., 2021*) and is a marker of acute severe lung distress (*Adcock et al., 2015*), an inflammatory condition that can lead to severe complications or death, further confirming the relevance of our model in mimicking the molecular events triggered by SARS-CoV-2 infection. Besides, in accordance with our data, a recent study showed that IMPDH2 mediates the activation of *CXCL8* expression and of NFkB transcriptional reporters during Nsp14 expression (*Li et al., 2021*).

Nevertheless, for a subgroup of genes the response appears to involve alterations of splicing patterns and other post-transcriptional events. Remarkably, splicing is strongly disrupted upon SARS-CoV-2 infection (*Banerjee et al., 2020*) and we see a significant overlap between the alternative splicing events induced by Nsp14 and the ones occurring during the infection. Furthermore, Nsp14 affects the expression of genes encoding proteins with functions in RNA processing and splicing. However, unlike the splicing modulation by Nsp16 which has been proposed to be general (*Banerjee et al., 2020*), the effect of Nsp14 seems to be specific to certain exons and introns as, in most cases, only one splicing event is modulated per gene. It is likely that Nsp16 and Nsp14 influence alternative splicing through different mechanisms, although we could not compare the effects of the two proteins as the plasmid for Nsp16 expression was not in the collection we utilized in our screening (*Gordon et al., 2020b*). In any case, while Nsp16 directly binds to pre-mRNA (*Banerjee et al., 2020*), Nsp14 is localized to the cytoplasm, so the latter modifies splicing indirectly, through a cascade of events that culminate in altered splicing patterns. Indeed, our experiments suggest that this might be through altered activity of IMPDH2, which has been shown to enter the nucleus and alter gene expression in other physiological contexts (*Kozhevnikova et al., 2012*; *Ahangari et al., 2021*). Altered splicing events could be a direct effect of IMPDH2 activity or secondary to the gene expression changes, as we showed that Nsp14 alters the expression of several RNA binding proteins and splicing regulators. As both Nsp14 and Nsp16 change the splicing patterns of several genes, further experiments will determine which is the exact contribution of Nsp14 and Nsp16 to the changes in alternative splicing following infection, as well as the cellular proteins mediating the effects of Nsp14.

Surprisingly, we also detected increased levels of circRNAs upon Nsp14 expression. Specifically, Nsp14 altered the expression of about 10% of the cellular circRNAs. The magnitude of this effect is striking, considering that deletion or knock-down of factors involved in circRNAs production generally

affects the expression of smaller percentages of circRNAs (*Conn et al., 2015*; *Errichelli et al., 2017*; *Knupp et al., 2021*). We showed that this increase is not due to enhanced transcription of the hosting RNAs, and our results indicate that Nsp14 influences biosynthesis and/or stability of those circRNAs. Published work showed that circRNAs accumulate in non-proliferating cells (*Bachmayr-Heyda et al., 2015*). Nsp14 alters the expression of genes regulating cell cycle and proliferation, and this may result in inhibition of pathways related to circRNA degradation or in inhibition of cell division, ultimately leading to circRNAs accumulation. However, this possible explanation will require future investigation.

Interestingly, recent studies indicate that circRNAs modulate some immune factors by repressing their pro-inflammatory activity in normal conditions (*Liu, 2019*). On the contrary, upon viral infection, circRNAs are degraded and these immune factors can orchestrate the appropriate response in the cell (*Liu, 2019*). Indeed, circRNA titer dramatically decreases upon viral infection (*Liu, 2019*; *Chen et al., 2020*). The fact that we detected an increase in levels of certain circRNAs upon Nsp14 expression suggests a possible mechanism for the immune evasion orchestrated by Nsp14.

Nsp14 is conserved among coronaviruses and it participates to viral replication and immune surveillance escape (*Ogando et al., 2020*). Indeed, on the one end it works as an ExoN, important for the correct replication of the viral genome, on the other hand it functions as N7-guanine-methyltransferase for modifying the 5' cap of the viral genome. Notably, both domains are important for successful viral replication and are considered potential drug targets (*Otava et al., 2021*; *Saramago et al., 2021*). Our data indicate that N7-guanine-methyltransferase domain, but not the ExoN one, is necessary for Nsp14 effects on host gene expression.

Nsp14 interacts with the replication complex and in proximity with other enzymes, such as Nsp16, involved in the modification of the 5' cap (*Romano et al., 2020*). Although we excluded the importance of the co-factor Nsp10 (*Ma et al., 2015*), we did not test how the combination of Nsp14 with other SARS-CoV-2 proteins modulate gene expression in the host cells. As mentioned above, in the future it would be interesting to co-express Nsp14 and Nsp16, since both are involved in the 5' cap modification (and, as mentioned before, the N7-guanine-methyltransferase domain of Nsp14 is important for the effects on gene expression), and both proteins disrupt splicing of cellular transcripts (*Banerjee et al., 2020*; *Romano et al., 2020*).

Finally, we showed that IMPDH2 mediates the effects on cell transcriptome, altered splicing, and circRNAs deregulation induced by Nsp14 expression, as pharmacological inhibition of IMPDH2 activity with MPA and MZR mitigated the effects of Nsp14. Intriguingly, MPA is a known immunosuppressant and antiviral agent (*Pan et al., 2012*; *To et al., 2016*; *Dang et al., 2017*), and it inhibits SARS-CoV-2 propagation (*Kato et al., 2020*).

## Ideas and speculations

Although our data clearly indicate that IMPDH2 mediates the effects induced by Nsp14, we still do not know the detailed mechanism. We reported altered expression of genes involved in purine synthesis, including a decrease in *IMPDH2* mRNA levels, and we also detected elevated cellular GTP concentrations upon Nsp14 expression. Taken together, these data suggest that Nsp14 activates IMPDH2. Besides, previous studies showed that Nsp14 interacts with IMPDH2 (*Gordon et al., 2020a*; *Gordon et al., 2020b*), but we do not know whether the direct interaction between Nsp14 and IMPDH2 is sufficient for triggering the massive gene expression changes or if this system requires a co-factor.

Notably, IMPDH2 binds nucleic acids (*McLean et al., 2004*; *Mortimer and Hedstrom, 2005*) and acts as a TF in *Drosophila* (*Kozhevnikova et al., 2012*). However, although IMPDH2 is conserved (*Hedstrom, 2009*) and can localize in the nucleus of mammalian cells (*Ahangari et al., 2021*), the TF activity of IMPDH2 has not been demonstrated in mammalian cells. Whether Nsp14 triggers cytoplasmic-nuclear shuffle of IMPDH2 in mammalian cells will require further investigation. Intriguingly, a recent study reported that IMPDH2 can bind a circRNA and that this interaction activates IMPDH2 enzymatic activity (*Wang et al., 2022*). In the future, it will be interesting to check the circRNA binding capacity of IMPDH2 in our model, where circRNAs expression is highly deregulated.

Besides, IMPDH2 can also associate with polyribosomes and modulates the translation of specific mRNAs (*Hsu, 2021*).

In addition, another recent report indicates that Nsp14 blocks the interferon response, which is upstream of the NFkB pathway, by shutting down global translation 24 hr after Nsp14 transfection (*Hsu, 2021*). However, our data indicate that the transcriptional activation of the NFkB pathway occurs very early after transfection. Furthermore, we confirmed that some effects are transcriptional, and we detected alterations on splicing and circRNAs levels, events which are not necessarily downstream to translational inhibition. Moreover, we showed that the transcriptional effects occur soon after Nsp14 expression, as treatment with MPA or MZR 8 hr after transfection partially recovered the gene expression profile of the cells. This indicates that some of the gene expression changes might precede the global translation shutoff (*Hsu, 2021*). Therefore, we speculate that Nsp14 can alter gene expression by disrupting both the transcriptome and translation. Another possibility is that the transcriptional and the translational effects depend on the levels of Nsp14, that cannot be compared between the two studies (*Hsu, 2021* and this study).

Although we showed that the N7-guanine-methyltransferase domain of Nsp14 is crucial for mediating the gene expression changes in the host cell, previous studies indicated that this domain is important for mediating Nsp14 translational repression (*Jin et al., 2013*; *Hsu, 2021*). This leads us to further speculate that Nsp14 may disrupt transcription, alternative splicing, and translation. Interestingly, Nsp14 can methylate free GTP in the cell and convert it to m7GTP, that is known to block cap-dependent translation by competing with mRNA for eIF4E (*Jin et al., 2013*). Our results indicate that Nsp14 increases cellular GTP concentration by activating IMPDH2. Therefore, a compelling model would be that Nsp14 activates IMPDH2, causing not only to altered transcription and splicing, but also to increased cellular GTP, that is then converted to m7GTP by Nsp14, ultimately leading to general translation repression. However, this hypothesis will require further investigation and could only explain only a small fraction of the changes observed when expressing Nsp14, as expression of a viral factor that inhibits protein synthesis directly (Nsp1) altered the levels of a much smaller set of genes.

To sum up, we here provided novel insights regarding the function of Nsp14 in altering gene expression of the host cell through the interaction with the cellular enzyme IMPDH2 in the context of SARS-CoV-2 infection.

## Materials and methods

**Key resources table**

| Reagent type (species) or resource | Designation | Source or reference | Identifiers | Additional information |
|---|---|---|---|---|
| Gene (*SARS-CoV-2*) | Nsp14 | wuhCor1/SARS-CoV-2 | NC_045512 v2:18040– 19620 | UCSC Genome Browser on SARS-CoV-2 Jan. 2020/NC_045512.2 Assembly (wuhCor1) |
| Gene (*Homo sapiens*) | IMPDH2 | UCSC Genome Browser on Human Feb. 2009 (GRCh37/ hg19) Assembly | NM_000884 | |
| Strain, strain background (*Escherichia coli*) | DH5alpha | Rosbash Lab | | Electrocompetent cells used for transformation of cloned plasmids |
| Cell line (*Homo sapiens*) | HEK293T | This paper | | Cell line obtained from Marr Lab Brandeis University. STR authenticated by ATCC, and mycoplasma tested. |
| Transfected construct (*SARS-CoV-2*) | pLXV-EF1alpha-2XStrep-SARS-CoV2-nsp14-IRES-Puro | Addgene | Plasmid #141380 | Human codon optimized |
| Transfected construct (*SARS-CoV-2*) | pLXV-EF1alpha-2XStrep-SARS-CoV2-nsp10-IRES-Puro | Addgene | Plasmid #141376 | Human codon optimized |
| Transfected construct (*jellyfish*) | pLXV-EF1alpha-2XStrep-eGFP-IRES-Puro | Addgene | Plasmid #141395 | Human codon optimized |

*Continued on next page*

*Continued*

| Reagent type (species) or resource | Designation | Source or reference | Identifiers | Additional information |
|---|---|---|---|---|
| Transfected construct (*SARS-CoV-2*) | pLXV-EF1alpha-2XStrep-SARS-CoV2-nsp14-D331A-IRES-Puro | This paper | | Human codon optimized, N7-guanine-methyltransferase mutant |
| Transfected construct (*SARS-CoV-2*) | pLXV-EF1alpha-2XStrep-SARS-CoV2-nsp14-D273A-IRES-Puro | This paper | | Human codon optimized, ExoN mutant |
| Transfected construct (*SARS-CoV-2*) | pLXV-EF1alpha-2XStrep-SARS-CoV2-nsp14-D90A/G92A-IRES-Puro | This paper | | Human codon optimized, ExoN mutant |
| Antibody | Anti-Strep (Mouse monoclonal) | Quiagen | Cat#: 34850 | WB (1:1000) |
| Antibody | Anti-IMPDH2 (Rabbit polyclonal) | Proteintec | Cat#: 12948-1-AP | WB (1:1000) |
| Antibody | Anti-Actin (Mouse monoclonal) | Cell Signaling | Cat#: 3700 | WB (1:1000) |
| Antibody | Anti-Nsp14 (Rabbit monoclonal) | Cell Signaling | Cat#: 99098 | WB (1:1000) |
| Antibody | Anti-Tubulin (Mouse monoclonal) | Sigma-Aldrich | Cat#: T5168 | WB (1:1000) |
| Antibody | Anti-Laminin B1 (Rabbit monoclonal) | Abcam | Cat#: ab133741 | WB (1:1000) |
| Antibody | Anti-GAPDH (Rabbit monoclonal) | Cell Signaling | Cat#: 2118 | WB (1:1000) |
| Antibody | Anti-Tri-Methyl-Histone H3 (Lys27) (Rabbit monoclonal) | Cell Signaling | Cat#: 9733 | WB (1:1000) |
| Antibody | Anti-mouse IgG- HRP conjugated (Rabbit monoclonal) | Millipore | Cat#: ap160p | WB (1:10,000) |
| Antibody | Anti-rabbit IgG HRP conjugated (Mouse monoclonal) | Millipore | Cat#: AP188P | WB (1:10,000) |
| Chemical compound, drug | Mizoribine IMPDH inhibitor | Selleckchem | Cat#: 50924-49-7 | Used at 10 µM, dissolved in DMSO |
| Chemical compound, drug | Mycophenolic acid (MPA) IMPDH inhibitor | Sigma-Aldrich | Cat#: M5255 | Used at 0.5 µM, dissolved in DMSO |
| Software, algorithm | GraphPad Prism software | GraphPad Prism (https://graphpad.com) | | RRID:SCR_002798 |
| Software, algorithm | R Project for Statistical Computing | R Project for Statistical Computing (https://www.r-project.org/) | | RRID:SCR_001905 |

## Experimental procedures

### Cell culture

HEK293T cells were a kind gift from Dr Marr at Brandeis University. HEK293T were authenticated by STR profiling through ATCC. HEK293T were maintained in Dulbecco's modified Eagle's medium (Genesee Scientific, 25–500), supplemented with 10% serum (FetalPlex, Gemini Bio/Fisher, # 100–602), and 1× penicillin-streptomycin (Penicillin-Streptomycin 100×, Genesee Scientific, 25-512). Mycoplasma test was performed using Mycoplasma Detection Kit by Southern Biotech (13100-01), according to manufacturer's instructions.

## Transient plasmid transfection

Plasmids pLVX-EF1alpha-SARS-CoV-2-proteins-2xStrep-IRES-Puro proteins are a gift from Nevan Krogan (Addgene) (*Gordon et al., 2020b*) are listed in *Supplementary file 3b* and purified by Midi-prep using Invitrogen kit. The day before transfection, $0.35*10^6$ cells per well were plated in 12-well plates. Then, cells were transfected with PEI (polyethylenimine 25 kD linear from Polysciences, cat# 23966-2). Briefly, 1500 ng of total DNA (1200 ng of pLVX-EF1alpha-SARS-CoV-2-proteins-2xStrep-IRES-Pur + 300 ng of pLVX-EF1alpha-eGFP-2xStrep-IRES-Puro) were incubated at RT for 20 min with PEI, with a ratio 3:1 = PEI:DNA in 100 µl of serum-free medium, and the mixture was added to the cells. Forty-eight hours after transfection, the medium was removed, and cells were harvested for following analysis.

## IMPDH2 inhibitors treatment

MPA (Sigma-Aldrich) was a kind gift from Hedstrom lab at Brandeis University and was reconstituted in DMSO (Sigma-Aldrich) and aliquots were stored at –20°C. Eight hours after transfection, MPA wash freshly diluted into cell media and cells were treated with a final concentration of 50 µM of MPA, or with an equivalent amount of DMSO.

MZR was purchased from Sellechem (50924-49-7), reconstituted in DMSO and aliquots were stored at −20°C. Eight hours after transfection, MZR wash freshly diluted into cell media and cells were treated with a final concentration of 10 µM of MZR, or with an equivalent amount of DMSO.

## RNA extraction and RT-qPCR

RNA was extracted with TriZol (Invitrogen) reagent following the manufacturer's instructions and then treated with DnaseI (NEB). For RT-qPCRs shown in *Figure 2F*, *Figure 2—figure supplement 2A*, and *Figure 4—figure supplement 1A*, cDNA was prepared with 1000 ng of DnaseI treated RNA (iScript Select cDNA Synthesis Kit, Bio-Rad, following manufacturer's instructions). Random hexamers were used for the nascent RNA experiment shown in *Figure 2F* and *Figure 2—figure supplement 2A*. OligodT primers were used for *Figure 4—figure supplement 1A*. The produced cDNA was utilized as a template for quantitative real-time PCR performed with the C1000 Thermal Cycler Bio-Rad. The PCR mixture contained Taq polymerase (SYBR green Bio-Rad). Cycling parameters were 95°C for 3 min, followed by 40 cycles of 95°C for 10 s, 55°C for 10 s, and 72°C for 30 s. Fluorescence intensities were plotted vs. the number of cycles by using an algorithm provided by the manufacturer. For the remaining RT-qPCRs, cDNA was synthetized using 1000 ng of RNA as input, 500 ng of oligodT (for mRNAs validation) or random hexamers (for circRNAs validation and alternative splicing validation), 50 mM Tris-HCl pH 8.3, 50 mM KCl, 3 mM $MgCl_2$, 10 mM DTT, 400 µM dATP, 400 µM dCTP, 400 µM dGTP, 400 µM dTTP, and homemade purified MLV-RT (200 units). The produced cDNA was utilized as a template for quantitative real-time PCR performed with the C1000 Thermal Cycler Bio-Rad. The PCR mixture contained homemade purified Taq polymerase, 20 mM Tris-HCl pH 8.8, 10 mM $(NH_4)_2SO_4$, 10 mM KCl, 2 mM MgSO4, 0.1% Triton X-100, 1× SYBR, 1.075 M ethylene glycol. Cycling parameters were 95°C for 3 min, followed by 40 cycles of 95°C for 10 s, 55°C for 30 s, and 72°C for 30 s. Fluorescence intensities were plotted vs. the number of cycles by using an algorithm provided by the manufacturer. All the primers used in this assay are listed in *Supplementary file 3c*. Regarding the RT-qPCR for detecting retained introns (*Figure 6G* and *Figure 6—figure supplement 2G*), we positioned one primer in the exonic sequence before or after the retained intron, and the other primer in the retained intronic sequence (indicated as exon-intron). In parallel, we also run a reaction using both primers in the exonic regions surrounding the retained intron. We normalized the abundance of the exon-intron and the exon-exon product on a housekeeping gene (*TBP* mRNA). Finally, we plotted the intronic-exonic ratio, defined as the ratio between the relative abundance of the exon-intron product on the relative abundance of the exon-exon one.

## Nascent RNA purification

Cells were harvested, washed with PBS, and then resuspended in 50 µl of Hypotonic Buffer (20 mM Tris-HCl pH 7.4, 10 mM KCl, 2 mM $MgCl_2$, 1 mM EGTA, 0.5 mM DTT, Murine RNAse Inhibitor [NEB], Complete Protease Inhibitor [Pierce, PIA32955]). Cells were incubated on ice for 3 min, then NP-40 to a final concentration of 0.1% was added, the cells were gently flicked, and incubated other 3 min in ice. Following, cells were centrifuged for 5 min at 4°C at 1000 rcf. The supernatant (corresponding

to the cytoplasmatic fraction) was collect in a separate tube and RNA was extracted with TriZol (Invitrogen), as indicated above. The nuclei in the pellet were gently resuspended in 250 µl of Homogenization Buffer (10 mM HEPES-KOH pH 7.5, 10 mM KCl, 1.5 mM MgCl$_2$, 0.8 M sucrose, 0.5 mM EDTA, 1 mM DTT, 0.5 unit/µl Murine RNase Inhibitor [NEB], protease inhibitor [Pierce, PIA32955]) were transferred onto 350 µl of sucrose cushion (10 mM HEPES-KOH pH 7.5, 10 mM KCl, 1.5 mM MgCl$_2$, 1 M sucrose, 10% glycerol, 0.5 mM EDTA, 1 mM DTT, 0.5 unit/µl Murine RNase Inhibitor [NEB], protease inhibitor [Pierce, PIA32955]), and centrifuged at 21,000 rcf for 10 min at 4°C. The supernatant was removed, and the nuclei (pellet) were resuspended in 150 µl of Nuclear Lysis Buffer (10 mM HEPES-KOH pH 7.5, 100 mM KCl, 10% glycerol, 0.1 M EDTA, 0.15 mM spermine [ThermoFisher], 0.5 mM spermidine [Sigma-Aldrich], 1 mM NaF, 0.1 mM Na$_3$VO$_4$, 0.1 mM ZnCl$_2$, 1 mM DTT, 1 unit/µl Murine RNase Inhibitor [NEB], protease inhibitor [Pierce, PIA32955]); 150 µl of 2× NUN Buffer (mM HEPES-KOH pH 7.5, 300 mM NaCl, 1 M urea, 1% NP-40, 1 mM DTT, 1 unit/µl Murine RNase Inhibitor [NEB], protease inhibitor [Pierce, PIA32955]) were added drop by drop, flicking the tube after each addition. Samples were incubated on ice for 20 min and centrifuged at 21,000 rcf for 30 min at 4°C. The supernatant (nucleoplasm) was collected, and RNA was extracted with TriZol LS (ThermoFisher), according to the manufacturer's protocol. Nascent RNA associated to chromatin (pellet) was extracted by adding 1 ml of TriZol, incubating the sample at 65°C for 15 min and then proceeding with the previously described extraction. All the fractions were treated with DNaseI (NEB), as previously described, before proceeding with RT-cDNA synthesis (iScript Select cDNA Synthesis Kit, Bio-Rad Random Priming, following manufacturer's instructions) and RT-qPCR (SYBR Green Bio-Rad), as indicated above.

## 3' RNA-seq libraries

Seventy-five ng of RNA were used as input for preparing 3' RNA sequencing libraries following CelSeq2 (*Hashimshony et al., 2016*) protocol, changing the UMI to six bases. Sequencing was performed on Illumina NextSeq 500 system.

## Total RNA-seq libraries

Forty ng of rRNA depleted RNA was used as a starting material for library preparation. NEBNext rRNA Depletion kit (NEB #E7405) was used to deplete ribosomal RNA. Libraries were prepared using the NEXTFLEX Rapid Directional RNA-Seq Kit 2.0 by PerkinElmer, following the manufacturer's instructions. The samples were sequenced by Novogene (Novogene Corporation Inc 8801 Folsom BLVD, Suite 290, Sacramento, CA) with Hiseq-4000.

## Metabolomic analysis

Cells were transfected as aforementioned, harvested with Trypsin-EDTA 0.25% 1× phenol red, Genesee Scientific (25-510) and washed three times with cold PBS. PBS was removed and samples were snap-frozen in liquid nitrogen. Samples were sent to Creative Proteomics 45-1 Ramsey Road, Shirley, NY, for metabolomic analysis. Briefly, cell pellet in each sample tube was made into 200 µl of 80% methanol. Cells were lysed on an MM 400 mill mixer with the aid of two metal balls at 30 Hz for 1 min three times, followed by sonication for 3 min in an ice-water bath. The samples were placed at - 20°C for 1 hr before centrifugal clarification at 21,000 *g* and 5°C for 15 min. The clear supernatant was collected for LC-MS and the precipitated pellet was used for protein assay using a standardized Bradford procedure. Standard solutions containing all targeted nucleotides were prepared, in a concentration range of 0.0002–20 nmol/ml, in an internal standard solution of 13C5-AMP, 13C10-ATP, and 13C10- GTP. Twenty µl of the clear supernatant of each sample was mixed with 180 µl of the internal standard solution. Ten µl aliquots of resultant sample solutions and standard solutions were injected to a C18 column (2.1 × 150 mm, 1.7 µm) to run UPLC-MRM/MS on a Waters Acquity UPLC system coupled to a Sciex QTRAP 6500 Plus mass spectrometer operated in the negative-ion ESI mode. The mobile phase was an acyl amine buffer (A) and acetonitrile/methanol (B) for binary gradient elution (5–65% B in 16 min), at 0.25 ml/min and 55°C. Concentrations of the detected analytes were calculated with internal-standard calibration by interpolating the constructed linear-regression curves of individual compounds, with the analyte-to-internal standard peak ratios measured from sample solutions.

## Protein extraction and Western blot

Cells were lysed in RIPA buffer (150 mM NaCl, 1% Triton X-100, 0.5% SDS, 50 mM Tris-HCl pH 8.0) supplemented with EDTA free protease inhibitor (Pierce, PIA32955) and centrifuged at 4°C 21.1 × 1000 $g$ for 30 min. Lysate was quantified by BCA Assay (ThermoFisher, 23225). Lysates were run on 4–20% gradient polyAcrylamide gels (Bio-Rad, 4568094) with Tris-Glycine SDS Running buffer. Transfer was performed in a Tris-Glycine buffer supplemented with 20% methanol. Membranes were blocked with 3% BSA in TBS-T Buffer pH 7.6 (Tris base 0.2 M, NaCl 1.5 M, 0.1% Tween-20). Membranes were incubated with the following primary antibodies: mouse anti-Actin (Cell Signaling, 3700), rabbit anti-IMPDH2 (Proteintec, 12948-1-AP), mouse anti-Strep-tag (Quiagen, 34850). Secondary antibodies: rabbit anti-mouse HRP conjugated (Millipore), mouse anti-rabbit HRP conjugated (Millipore). Detection was performed using ECL (Pierce, 32106).

## Subcellular fractionation and chromatin precipitation

Cells were harvested, washed with PBS, and then resuspended in Hypotonic Buffer (20 mM Tris-HCl pH 7.4, 10 mM KCl, 2 mM $MgCl_2$, 1 mM EGTA, 0.5 mM DTT, Murine RNAse Inhibitor [NEB], Complete Protease Inhibitor [Pierce, PIA32955]). Cells were incubated on ice for 3 min, then NP-40 to a final concentration of 0.1% was added, the cells were gently flicked, and incubated other 3 min in ice. Following, cells were centrifuged for 5 min at 4°C at 1000 rcf. The supernatant (corresponding to the cytoplasmatic fraction) was collect in a separate tube. The nuclei in the pellet were gently washed with PBS and then resuspended in 250 µl of Homogenization Buffer (10 mM HEPES-KOH pH 7.5, 10 mM KCl, 1.5 mM $MgCl_2$, 0.8 M sucrose, 0.5 mM EDTA, 1 mM DTT, 0.5 unit/µl Murine RNase Inhibitor [NEB], protease inhibitor [Pierce, PIA32955]) were transferred onto 350 µl of sucrose cushion (10 mM HEPES-KOH pH 7.5, 10 mM KCl, 1.5 mM $MgCl_2$, 1 M sucrose, 10% glycerol, 0.5 mM EDTA, 1 mM DTT, 0.5 unit/µl Murine RNase Inhibitor [NEB], protease inhibitor [Pierce, PIA32955]), and centrifuged at 21,000 rcf for 10 min at 4°C. The supernatant was removed, and the nuclei (pellet) were washed in PBS and then resuspended in 20 µl of Nuclear Lysis Buffer (10 mM HEPES-KOH pH 7.5, 100 mM KCl, 10% glycerol, 0.1 M EDTA, 0.15 mM Spermine [ThermoFisher], 0.5 mM Spermidine [Sigma-Aldrich], 1 mM NaF, 0.1 mM $Na_3VO_4$, 0.1 mM $ZnCl_2$, 1 mM DTT, 1 unit/µl Murine RNase Inhibitor [NEB], protease inhibitor [Pierce, PIA32955]). Twenty µl of 2× NUN Buffer (mM HEPES-KOH pH 7.5, 300 mM NaCl, 1 M urea, 1% NP-40, 1 mM DTT, 1 unit/µl Murine RNase Inhibitor [NEB], protease inhibitor [Pierce, PIA32955]) were added drop by drop, flicking the tube after each addition. Samples were incubated on ice for 20 min and centrifuged at 21,000 rcf for 30 min at 4°C. The supernatant (nucleoplasm) was collected and transferred to a separate tube. Proteins associated to chromatin (pellet) were treated with DNaseI (NEB). Finally, we proceeded with SDS-PAGE and Western blot (loaded the entire collected fractions), as described above.

## Cloning of Nsp14 ExoN mutants

### pLXV-EIF1 alpha-2xStrep-SARS-CoV-2-nsp14-D90A/G92A-IRES-Puro

pLXV-EF1alpha-2xStrep-SARS-CoV-2-nsp14-IRES-Puro was opened with BsrGI-HF (NEB) and EcoRI-HF (NEB). The removed cassette was replaced by ligation with a G-block ordered from GenScript (GenScript UAS Inc, Piscataway, NJ), previously amplified with specific primers.

G-block:

Gaattcgccgccaccatgtggtccatccgcagtttgagaaagggtggtggttcaggcggaggctccggggg cgggagctggtctcaccgcaatttgaaaaaggcgctgcggctgctgaaaatgtaacgggcttgtttaaaga ctgtagtaaagtgatcacaggactccacccacacaagcacctacccacctttccgtagatacgaagttcaaaa cggaaggattgtgtgtggatataccaggatgatccaaaggatatgacgtaccgaaggctgatttccatgatgggg tttttaagatgaattaccaagttaatggctaccccaacatgttcatcaccaggggagggggcaattagacacgtaa gagcctggtaggcttcGCCgttGCCggttgccatgcaaccccgggaagccgtaggcacaaaccttccgttg cagcttggcttttccacggggcgtcaacctcgttgccgtaccgactggctatgttgacacgccgaacaacaccga tttctctcgcgtaagtgctaagcctcctccgggagatcaattcaagcatcttatacctctcatgtaca.

Primers:

D90AG92A_F: cacacgaattcgccgccac.

D90AG92A _R: cacactgtacatgagaggtataaga.

## pLXV-EIF1 alpha-2xStrep-SARS-CoV-2-nsp14-D273A-IRES-Puro

pLXV-EF1alpha-2xStrep-SARS-CoV-2-nsp14-IRES-Puro was opened with BstBI (NEB) and AfeI (NEB). The removed cassette was replaced by ligation with a G-block ordered from GenScript (GenScript UAS Inc, Piscataway, NJ), previously amplified with specific primers.

G-block:
ccac actt cgaa ctta cttc tatg aaat actt tgta aaaa tcgg cccc gagc ggac atgt tgtt tgt gcga ccga cgag ctac ttgt tttta gcac agca tctg acac ctac gcat gctg gcac caca gtat aggc ttcg att acgt ctac aatc cctt tatg atag acgt acaa caat gggg cttt acgg gtaa cttg caga gtaa tcac gatc ttta ctgc caag ttca tggg aacg caca cgtg gcct cctg cGCC gcga taat gacg aggt gctt ggcc gtgc acga g tgct ttgt taag cggg tcga ttgg acta taga gtat ccca taat cggt gacg aact taaa atta atgc tgca tgca ggaaagtgcagcacatggtagtaaaagcagcgct.

Primers:

D273A_F: ccacacttcgaacttacttctatg
D273A_R: cacacagcgctgcttttactacc

## Cloning of Nsp14 N7-guanine-methyltransferase mutant

## pLXV-EIF1 alpha-2xStrep-SARS-CoV-2-nsp14-D331A-IRES-Puro

pLXV-EF1alpha-2xStrep-SARS-CoV-2-nsp14-IRES-Puro was opened with NdeI (NEB) and AfeI (NEB). The removed cassette was replaced by ligation with a G-block ordered from GenScript (GenScript UAS Inc, Piscataway, NJ), previously amplified with specific primers.

G-block:
agcg ctcc tcgc ggat aagt ttcc tgtg ctcc acGC Catc ggca accc taag gcca ttaa a tgcgtcccccaagctgatgtagagtggaaattctatgacgctcagccatgtagtgacaaagcatacaagattgaagaattgttttattcata tgcacacc

Primers:

D331A_F: cacacagcgctcctcgcg
D331A_R: ggtgtgcatatgaataaaacaattc

## Cloning of CXCL8-Firefly reporter

pGL_RSV_RF_BG (a modification of the pGL plasmid) was a kind gift from Dr Marr at Brandeis University. pGL_RSV_RF_BG was opened with XhoI and NcoI. DNA was extracted from HEK293T cells with DNeasy Blood & Tissue Kit Quiagen (ID: 69504) according to the manufacturer's instructions. The CXCL8 promoter was amplified by PCR with Phusion High-Fidelity PCR Master Mix with HF Buffer (NEB) according to the manufacturer's instruction with the following primers:

CXCL8_promoter_R:     CACCACCATGGTGGCTAGCAGCTGGTACCCAGCTTGGACCTGGC TCTTGTCCTAGAA
CXCL8_promoter_F: CACCACTCGAGACAGTTCCTAGAAACTCTCTAAAATGCTTAGAA

The purified insert was digested with XhoI and NcoI and inserted into the plasmid by ligation with T7 DNA ligase (NEB) following the manufacturer's instructions to generate the pGL_CXCL8_promoter_Firefly.

## Luciferase assay

The day before transfection 0.18*106 cells per well were plated into 24-well plates. Cells were transfected as aforementioned using 750 ng of total DNA. Cells were transfected with 75 ng of Firefly reporter, 75 ng of Renilla reporter, and 600 ng of pLVX-EF1alpha-SARS-CoV-2-Nsp14-2xStrep-IRES-Puro or 600 ng of empty vector. We used the following plasmids as reporters: pGL_CXCL8_promoter_Firefly, generated as indicated above.

pHAGE NFkB-TA-LUC-UBC-GFP-W was a gift from Darrell Kotton (Addgene plasmid # 49343; http://n2t.net/addgene:49343; RRID:Addgene_49343) (*Wilson et al., 2013*).

7xE-Box::Renilla was a gift from Koen Venken (Addgene plasmid # 124532; http://n2t.net/addgene: 124532; RRID:Addgene_124532, *Sarrion-Perdigones et al., 2020*).

Forty-eight hours after transfection, cells were lysate in lysis buffer (25 mM Tris-phosphate at pH 7.8, 10% glycerol, 1% Triton X-100, 1 mg/ml of bovine serum albumin [BSA], 2 mM cyclohexylene diamin tetraacetate, and 2 mM DTT). An aliquot of the lysate was added to Firefly Buffer 1× (75 mM Tris pH 8.0, 100 µM EDTA, 530 µM ATP, 5 mM MgSO$_4$) freshly supplemented with 0.1 µM D-luciferin (Goldbio.com) and 100 µM Coenzyme-A (Fisher, #BP2510250), and luminescence was measured. Immediately after, an equal amount of Renilla Buffer 1× supplemented with 10 µM Coelenterazine ( Goldbio.com) was added to the sample and luminescence was measured again. Non-transfected cell lysate was used to measure the background signal, that was subtracted before calculating the Firefly/ Renilla ratios.

## Bioinformatic analyses

### Linear RNA alignment and quantification

Raw reads were aligned to the human genome (hg38) using STAR (*Dobin et al., 2013*). Mapped reads in the 3' UTR were quantified using End Sequence Analysis Toolkit (*Gohr and Irimia, 2019*) for 3' RNA libraries. FeatureCounts (*Gohr and Irimia, 2019*) was used to quantify mapped transcripts from total RNA-seq libraries. DEGs were found using DESeq2 as previously described (*Sarrion-Perdigones et al., 2020*). DEGs with |L2FC| > 1, p-adjusted value <0.05 were considered significant and used as input in downstream GO analysis (DAVID, v 6.8). For the analysis of the time point 3' RNA-seq experiment, we used |L2FC| > 0.5, p-adjusted value <0.05. HEK293T cell transcriptome was used as a reference to query for enriched GO terms from up- and downregulated DEG lists. GSEA was performed using the fgsea package in R (*Sergushichev et al., 2016*). Gene rank was determined prior to GSEA by calculating −log(p-value)* sign(log2 fold change)(*Sergushichev et al., 2016*). Gene sets used for GSEA were downloaded from Molecular Signatures Database (http://www.msigdb.org). To compare our RNA-seq results with other studies, we used a gene set from MSigDB containing DEGs from published RNA-seq data. A curated list of terms from GSEA were plotted to assess the transcriptional response to viral protein expression. Data visualization was carried out using ggplot2 in R. Non-omics statistical analysis and data visualization was completed using Prism.

### circRNA detection and differential expression analysis

Computational analysis of circRNAs was performed on total RNA-seq data. circRNAs were detected by searching unaligned reads for head-to-tail splice junctions as previously described (*Memczak et al., 2013*). Differentially expressed circRNAs were found by DESeq2 (*Love et al., 2014*). circRNA reads were normalized using size factors computed by DESeq2 together with all mapped reads.

### Intronic reads quantification and analysis

Reads were aligned with STAR-aligner (*Wilson et al., 2013*) to the human genome and transcriptome version Hg38. Intronic reads were extracted and counted using Featurecounts function in R with an intronic region reference.

### Alternative splicing analysis

Percentage of inclusion (PSI) quantification was done with Vast-Tools (*Tapial et al., 2017*). Delta PSI was then calculated from the mean of each condition. To determine events changing between Nsp14 and transfection control, we chose a minimum of 15% difference in mean PSI and no overlapping between replicas. Introns and exons feature analysis was done using Matt (*Gohr and Irimia, 2019*).

With the same approach, we re-analyzed total RNA-seq from HEK293T-hACE2 cells infected with SARS-CoV-2 GSE169158 (*Sun et al., 2021*). Then, we tested the significance of the overlap between the alternative splicing analysis on our dataset and this dataset by performing a randomization of sets with equal size, evaluating their overlap to generate a null distribution, and calculating the probability of our overlap (10,000 replicas p-value < 0.01).

## Motif enrichment analysis

We extracted the promoters' sequences of upregulated genes with a fold change of at least 1.75 (log2FC > 0.8 and adjusted p-value > 0.05) using windows of 100, 500, and 1000 bases upstream of each annotated transcription start site for each gene. We then used the MEMEsuite (*Bailey et al., 2009*) using the tool AME (*McLeay and Bailey, 2010*) for motif enrichment analysis. We used the motif data base *Homo sapiens* COmprehensive MOdel COllection (HOCOMOCO) v11 TF binding models (binding profiles or binding motifs) for human TFs. As control for the input sequences, we used the tool provided for scrambled sequences option. Finally, we used MEMEsuit tool FIMO (*Grant et al., 2011*) to identify the position of these motifs in the analyzed 100 bp promoters. We extracted the promoters' sequences of upregulated genes with a fold change of at least 1.75 (log2FC > 0.8 and adjusted p-value > 0.05) using windows of 100, 500, and 1000 bases upstream of each annotated transcription start site for each gene. We then used the MEMEsuite tool AME (*McLeay and Bailey, 2010*) for motif enrichment analysis using. We used the motif data base HOmo sapiens COmprehensive MOdel COllection (HOCOMOCO) v11 TF binding models (binding profiles or binding motifs) for human TFs. As control for the input sequences, we used the tool provided scrambled sequences option. Finally. we used MEMEsuit tool FIMO (*Grant et al., 2011*) to identify the position of these motifs in the analyzed 100 bp promoters.

## Acknowledgements

We thank the Hedstrom Lab, the Marr Lab, and the Rosbash Lab at Brandeis University for sharing reagents, instruments, and for helpful discussion. Thanks to Dr Shifman at the Hebrew University of Jerusalem for help and suggestions with the bioinformatic analysis. Thanks Dr tenOever at Icahn School of Medicine at Mount Sinai for critical discussion of the data. Thanks to Dr Aarti Jajoo for the initial steps of the screening data analysis, and to Claire Martel for helping with plasmids purification. Special appreciation to Valentina Tacconi for preparing the Potential Striking Image. We warmly thank all the members of Kadener lab for helpful and critical discussion.

## Additional information

### Funding

No external funding was received for this work.

### Author contributions

Michela Zaffagni, Conceptualization, Data curation, Formal analysis, Investigation, Methodology, Project administration, Validation, Visualization, Writing - original draft, Writing - review and editing; Jenna M Harris, Conceptualization, Data curation, Formal analysis, Investigation, Methodology, Software, Writing - review and editing; Ines L Patop, Conceptualization, Data curation, Formal analysis, Investigation, Methodology, Software, Visualization, Writing - original draft, Writing - review and editing; Nagarjuna Reddy Pamudurti, Data curation, Validation; Sinead Nguyen, Data curation, Methodology, Resources, Validation; Sebastian Kadener, Conceptualization, Funding acquisition, Project administration, Supervision, Writing - original draft, Writing - review and editing

### Author ORCIDs

Michela Zaffagni (ID) http://orcid.org/0000-0002-2483-8727
Sebastian Kadener (ID) http://orcid.org/0000-0003-0080-5987

### Decision letter and Author response

Decision letter https://doi.org/10.7554/eLife.71945.sa1
Author response https://doi.org/10.7554/eLife.71945.sa2

## Additional files

### Supplementary files

• Supplementary file 1. *Supplementary file 1a*: List of differentially expressed genes (DEGs) in the screening described in *Figure 1*. 3' RNA-seq. *Supplementary file 1b*: List of Gene Ontology (GO) terms described in *Figure 1*. 3' RNA-seq. *Supplementary file 1c*: Gene set enrichment analysis (GSEA) described in *Figure 2*. Total RNA-seq. *Supplementary file 1d*: Differential gene expression results described in *Figure 2*. *Supplementary file 1e*: Alternative splicing analysis results described in *Figure 3*. *Supplementary file 1f*: Feature analysis with MATT for upregulated intron retentions and downregulated exon inclusion events described in *Figure 3—figure supplement 1*. *Supplementary file 1g*: Alternative splicing analysis on a published total RNA-seq dataset of HEK293T-hACE2 cells infected with severe acute respiratory syndrome coronavirus 2 (SARS-CoV-2) (*Sun et al., 2021*). *Supplementary file 1h*: Deregulated circRNAs described in *Figure 3*.

• Supplementary file 2. *Supplementary file 2a*: Differentially expressed gene (DEG) at different time points described in *Figure 5*. 3' RNA-seq. *Supplementary file 2b*: Gene Ontology (GO) described in *Figure 5*. 3' RNA-seq. *Supplementary file 2c*: AME analysis with 100 bp of promoter described in *Figure 5*. *Supplementary file 2d*: FIMO analysis with 100 bp promoter described in *Figure 5*. *Supplementary file 2e*: AME analysis with 1000 bp promoter described in *Figure 5*.

• Supplementary file 3. *Supplementary file 3a*: Concentrations of cellular metabolites detected by HPLC/MS. *Supplementary file 3b*: List of plasmids used for severe acute respiratory syndrome coronavirus 2 (SARS-CoV-2) protein expression in the initial screening described in *Figure 1*. *Supplementary file 3c*: List of primers used for the RT-qPCR assays.

• Transparent reporting form

### Data availability

1.RNAseq data generated in this study is in GEO (GSE179251). 2.RNA seq data already published and re-analyzed in this study are the following: -Sun, G., Cui, Q., Garcia, G. et al. Comparative transcriptomic analysis of SARS-CoV-2 infected cell model systems reveals differential innate immune responses. Sci Rep 11, 17146 (2021). https://doi.org/10.1038/s41598-021-96462-w, GSE169158 -Blanco-Melo D, Nilsson-Payant BE, Liu WC, Uhl S, Hoagland D, Møller R, Jordan TX, Oishi K, Panis M, Sachs D, Wang TT, Schwartz RE, Lim JK, Albrecht RA, tenOever BR. Imbalanced Host Response to SARS-CoV-2 Drives Development of COVID-19. Cell. 2020 May 28;181(5):1036-1045.e9. doi: 10.1016/j.cell.2020.04.026. Epub 2020 May 15. PMID: 32416070; PMCID: PMC7227586, GSE147507 All data generated or analyzed during this study are included in the manuscript and supporting files.

The following dataset was generated:

| Author(s) | Year | Dataset title | Dataset URL | Database and Identifier |
|---|---|---|---|---|
| Kadener S, Zaffagni M, Harris JM, Patop IL, Nagarjuna RP, Sinead N | 2022 | SARS-CoV-2 Nsp14 mediates the effects of viral infection on the host cell transcriptome | https://www.ncbi. nlm.nih.gov/geo/ query.cgi?acc= GSE179251 | NCBI Gene Expression Omnibus, GSE179251 |

The following previously published datasets were used:

| Author(s) | Year | Dataset title | Dataset URL | Database and Identifier |
|---|---|---|---|---|
| Sun G, Cui Q, Wang C, Garcia G | 2021 | Comparative transcriptomic analysis of SARS-CoV-2 infected cell model systems reveals differential innate immune responses | https://www.ncbi. nlm.nih.gov/geo/ query.cgi?acc= GSE169158 | NCBI Gene Expression Omnibus, GSE169158 |
| Blanco-Melo D, Nilsson-Payant BE, Liu WC, Uhl S, Hoagland D, Møller R, Jordan TX, Oishi K, Panis M, Sachs D, Wang TT, Schwartz RE, Lim JK, Albrecht RA, tenOever BR | 2020 | Imbalanced Host Response to SARS-CoV-2 Drives Development of COVID-19 | https://www.ncbi. nlm.nih.gov/geo/ query.cgi?acc= GSE147507 | NCBI Gene Expression Omnibus, GSE147507 |

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
