## [Editor Report]

The paper shows that expression of the SARS-CoV-2 Nsp14 protein, which is involved in viral RNA replication, provokes a transcriptional profile that strongly resembles that observed following SARS-CoV-2 infection. Moreover, Nsp14 expression alters the splicing of many genes, increases the number of circRNAs, and activates the NFkB pathway. This surprising observation gives new insight into the biology of SARS-CoV-2 and may have implications for therapy.

---

## [Decision Letter]

**Decision letter after peer review:**

Thank you for submitting your article "SARS-CoV-2 Nsp14 mediates the effects of viral infection on the host cell transcriptome." for consideration by *eLife*. Your article has been reviewed by 2 peer reviewers, and the evaluation has been overseen by Kevin Struhl as the Senior and Reviewing Editor. The following individuals involved in review of your submission have agreed to reveal their identity: Hyerim Yi (Reviewer #1); Ryan A Flynn (Reviewer #2).

Essential revisions:

1) Role of Nsp14 function in authentic infection

2) Rescue experiment for IMPDH2.

In addition, the issue related to Nsp14 expression level in a time course experiment should be verified (Figure 5).

A weakness of this paper is lack of connection among their observations that are described in parallel. They are arguing that IMPDH2 is a key mediator of the Nsp14 effect on the host transcriptome, but they only validated the effect of IMPDH2 on the differential gene expression. Since they have already generated the data comparing Nsp14+MPA to Nsp14+DMSO (Figure 6E), it will be worth analyzing the IMPDH2 effect on splicing or circRNA expression that are described here.

*Reviewer #1 (Recommendations for the authors):*

1. How would the data look like when genes under the Nsp14-mediated alternative splicing are presented in Figure 2A together?

2. Please show less processed data such as scatter or MA plots for Figure 6E.

*Reviewer #2 (Recommendations for the authors):*

A major question I have after reading your work is: where is Nsp14?

Subcellular imaging:

– Where is Nsp14 in human cells during native infections?

– Where is Nsp14 in human cells during individual transfection?

– This would be additionally interesting in the experimental set up of Figure 5 where changes in the DEGs evolve over time and correlating where Nsp14 is with particular sets of genes could be very revealing.

Whole cell protein levels:

– How much higher, per cell, is the transfection of Nsp14 than what accumulates during native infections?

– Again, here, for Figure 5, it would be very important to know WHEN Nsp14 is actually expressed/present in the cell. There are not many changes after 8 hours, but how much Nsp14 have the 293 cells even made at this point?

The fact that "more than 50% of the changes in the transcriptome upon Nsp14 expression are transcriptional" seems like it warrants a bit of investigation and hopefully some imaging or subcellular fractionation at different time points could at least provide a framework to propose hypotheses for how this is possible in the context of the other mechanisms you're proposing

– One could also do a chromatin-fractionation to see if Nsp14 sticks to chromatin, better / more complete to do ChIP-seq but this could be overkill for this study

The circRNA changes are interesting at a high level however it is hard to know how surprising or unique this change is; both that it changes and the magnitudes (number of circRNAs changing). Are there data that this could be compared to? For example, KD, KO, or OE of other splicing factors in experimental or disease conditions that show fewer, similar, or more changes in circRNAs, and importantly how these numbers compare to the changes in the linear mRNA pool.

The experimental idea in Figure 4, with co-expression is quite nice however limited in scope. This may be ok for this study, but it is hard to know how all these factors work together during a native infection when studying them in isolation. It's convenient that Nsp14 has such a singularly robust impact of the host transcriptome, but one wonders how the combination of the other gene products would better tune or in other ways influence the host cell's state.

[Editors' note: further revisions were suggested prior to acceptance, as described below.]

Thank you for resubmitting your article "SARS-CoV-2 Nsp14 mediates the effects of viral infection on the host cell transcriptome" for consideration by *eLife*. Your revised article has been reviewed by 2 peer reviewers, and the evaluation has been overseen by Kevin Struhl as the Senior and Reviewing Editor. The following individuals involved in review of your submission have agreed to reveal their identity: Hyerim Yi (Reviewer #1); Ryan A Flynn (Reviewer #2).

The paper shows that expression of the SARS-CoV-2 Nsp14 protein, which is involved in viral RNA replication, provokes a transcriptional profile that strongly resembles that observed following SARS-CoV-2 infection. Moreover, Nsp14 expression alters the splicing of many genes, increases the number of circRNAs, and activates the NFkB pathway. This surprising observation gives new insight into the biology of SARS-CoV-2 and may have implications for therapy.

*Reviewer #1 (Recommendations for the authors):*

The authors addressed most of the points raised by reviewers and enhanced the manuscript with compelling data. Although they did not address the comment regarding the role of Nsp14 function in authentic infection, I think that the function of Nsp14 observed upon exogenous expression is well addressed. I do understand the limitation of doing BSL3 experiment. But here I have some questions and suggestions related to their response to this comment.

1. The authors showed the cytoplasmic localization of exogenous Nsp14 as one of the supporting evidence of resemblance to SARS-COVID-2 infection situation. I wonder how cytoplasmic localization of Nsp14 could be associated with altered splicing events which occur in the nucleus. It would be nice to include any speculation on this in the discussion.

2. To further support the role of Nsp14 in altered splicing upon the real SARS-CoV-2 infection, I would suggest reinforcing the specificity of Nsp14 on altered splicing. Given that Nsp16 is also known for altered splicing, especially causing intron retention which is the common phenomenon the authors observed with Nsp14, I still wonder how much Nsp14 would explain the altered splicing upon SARS-COVID-2 infection.

*Reviewer #2 (Recommendations for the authors):*

The authors carefully read the initial reviews, extended their work, and I think addressed the major of the conceptual points raised. The additional data and controls included provide better context in some cases and strengthen the mechanistic claims overall. Parts of the news results and Discussion sections I think also better contextualize some of the initial ideas that were less clearly stated in the first draft. As far as I am concerned, I think this manuscript should proceed to publication.

---

## [Author Response]

Essential revisions:1) Role of Nsp14 function in authentic infection

We thank the reviewers for bringing this up. We have now addressed this in the new version of the manuscript by adding several experiments and changing text.

In the previous version of the manuscript, we reported the surprising finding that expression of Nsp14 recapitulates an important part of the transcriptome changes observed following SARS-CoV-2 infection. We have now added additional experiments and analysis to build up on those similarities. More specifically:

We show that Nsp14 localizes in the cytoplasm, similarly to what happens during infection. These results are in the new Figure 6—figure supplement 1A.We show that at least at the RNA level, the amounts of Nsp14 are comparable between our experiments and infection (See comment 2 to reviewer 2)During SARS-CoV-2 infection many genes showed altered splicing. Previous research suggested that this is mostly due to Nsp16, that interacts with U1 and U2. However, we also detected alteration of splicing when we express Nsp14, and we showed a significant overlap between the altered splicing events upon infection and Nsp14 expression (about 10% of the events, see Supplementary File 1 and comment 1 to reviewer 1). This suggests that Nsp14 expression contributes also to the alternative splicing events that occur during the infection, although we were not able to dissect the contribution of Nsp14 and Nsp16.Regarding determining whether the changes in circRNA levels are also observed during infection, we found that this comparison is not possible because circRNA levels are known to globally decrease following viral infection (personal communication Kadener lab, see comment 1 to reviewer 1)

Furthermore, we have extended the explanation of the similitude between the transcriptional changes observed upon Nsp14 expression in our experimental model and physiological viral infection, and we explained better in the Introduction and Discussion the role of Nsp14 during infection.

We recognize the limitation of our model in which Nsp14 might be expressed at higher levels than during the physiological infection (despite our observation at the RNA level, see answer to reviewer 2). Nevertheless, similar expression approaches have been used to study the function of other SARS-CoV-2 proteins and explained molecular mechanisms that occur during the infection. Furthermore, we showed that our model recapitulates the gene expression changes, as well as some of the alternative splicing events that occur during SARS-CoV-2 infection.

2) Rescue experiment for IMPDH2.

We thank the reviewers for bringing this up. To address this issue, we have extended the rescue experiment for IMPDH2 by:

Demonstrating that the effects of Nsp14 on circRNA expression is partially rescued by inhibition of IMPDH2 by MPA (Figure 6—figure supplement 2C). Furthermore, we showed that MPA treatment partially rescues also the tested alternative splicing events (Figure 6G).We recapitulated the partial rescue of on mRNA, circRNAs, and alternative splicing upon MPA treatment by utilizing a second IMPDH2 inhibitor (Mizoribine) (Figure 6G and Figure 6-figure supplement 2E, 2F, 2G).

In addition, the issue related to Nsp14 expression level in a time course experiment should be verified (Figure 5).

We understand the concern raised by the reviewers regarding the expression levels of Nsp14 at early time points during the time course experiment described in Figure 5. We repeated the transfection, and we determined Nsp14 levels by Western Blot. We detected Nsp14 protein expression as early as 6h post transfection, and, as expected, Nsp14 levels increased over time (see Figure 5—figure supplement 1B). This is consistent with the results from the 3’RNAseq experiment and it demonstrates that the observed early response is mediated by Nsp14 protein. We thank the reviewer for bringing this up.

Besides, we checked also the amount of normalized *Nsp14* reads in our 3’ RNA seq dataset (see Figure 5—figure supplement 1A). As expected, we detected increasing amount of *Nsp14* normalized reads through our time course (less than 5,000 reads 6h after transfection and more than 15,000 at 48h).

A weakness of this paper is lack of connection among their observations that are described in parallel. They are arguing that IMPDH2 is a key mediator of the Nsp14 effect on the host transcriptome, but they only validated the effect of IMPDH2 on the differential gene expression. Since they have already generated the data comparing Nsp14+MPA to Nsp14+DMSO (Figure 6E), it will be worth analyzing the IMPDH2 effect on splicing or circRNA expression that are described here.

We thank the reviewers and the editor for their helpful suggestions, we believe that the experiments added now contributed to make the manuscript tighter and more connected. Following this last suggestion, we now added an experiment in which we showed by RT-qPCR that treatment with two IMPDH2 inhibitors (MPA and Mizoribine) partially rescue intron retention, mRNAs, and circRNAs expression (see Figure 6G, Figure 6—figure supplement 2C, E, F, G). Unfortunately, the generated dataset explained in Figure 6 is a 3’ RNA seq experiment, not suitable for circRNAs detection and alternative splicing analysis, that’s why we now utilized RT-qPCR.

Reviewer #1 (Recommendations for the authors):1. How would the data look like when genes under the Nsp14-mediated alternative splicing are presented in Figure 2A together?

We thank the reviewer for bringing up this point. The expression of most of the genes that show alternative splicing events is not deregulated upon Nsp14 expression. We have now presented the requested plot in Figure 3—figure supplement 1B.

2. Please show less processed data such as scatter or MA plots for Figure 6E.

We thank the reviewer for pointing this out. We have now presented the data as suggested in Figure 6—figure supplement 1B.

Reviewer #2 (Recommendations for the authors):A major question I have after reading your work is: where is Nsp14?Subcellular imaging:– Where is Nsp14 in human cells during native infections?– Where is Nsp14 in human cells during individual transfection?– This would be additionally interesting in the experimental set up of Figure 5 where changes in the DEGs evolve over time and correlating where Nsp14 is with particular sets of genes could be very revealing.

We thank the reviewer for raising these interesting and important points. Nsp14 is part of the Replication Complex. Recent research showed that, as for other coronaviruses, replication of SARS-CoV-2 genome takes place in replication organelles that provide a protective environment for the newly synthesized viral genome(V’kovski et al. 2021). Interestingly, these organelles are formed in the cytoplasm and present convoluted double layered membranes. A recent EM study showed that these organelles have pores and that likely exchange material with the cytoplasm(Wolff et al. 2020).

A previous study using the same plasmid we utilized showed that Nsp14 is cytoplasmatic by immunohistochemistry (Gordon, Hiatt, et al. 2020). In any case, we now determined the localization using a complementary approach and performing subcellular fractionation and chromatin isolation at different timepoints after transfection (12h, 24h. 48h). In agreement with previous publication, our results showed that Nsp14 is exclusively cytoplasmic (new Figure 6 – —figure supplement 1A). This strongly indicates that the transcriptional changes induced by Nsp14 are mediated by other cellular factors.

Whole cell protein levels:– How much higher, per cell, is the transfection of Nsp14 than what accumulates during native infections?

We understand the point raised by the reviewer. To partially address this issue, we have compared the levels of the RNA encoding Nsp14 during the infection to the ones obtained following transfection and showed they are comparable (indeed higher during the infection; see (Author response image 1) ). Specifically, in our total RNA seq we detect about 10^5^ normalized reads for *Nsp14* and in the dataset of total RNA seq from HEK293T-hACE2 infected with SARS-CoV-2(Sun et al. 2021) we detected more than 10^6^ normalized reads for *Orf1ab* (Nsp14 is generated from proteolytic cleavage of polyprotein Orf1ab) (see Author response image 1).

**Author response image 1. sa2fig1:** 

We are aware that this is just an approximative comparison since many RNA molecules are not translated but packaged into new viral particles during the infection. We are aware about the limitations of this experiment as it doesn’t determine protein levels and we have stated this clearly in the new version of the manuscript. Nevertheless, we believe that, although the expression levels of Nsp14 are likely higher in the transfection experiments, our results are still very relevant given the strong overlap observed with the gene expression transcriptional profile. Furthermore, in this new version of the manuscript, we re-analyzed a published dataset of infected cells and we showed significant overlap between alternative splicing events that occur during the infection and upon Nsp14 expression, further indicating that our model partially recapitulates what is happing during the infection. Last but not least, our study shows that the effects of Nsp14 are mediated by IMPDH2 (or at least require IMPDH2 activity) because pharmacological inhibitors of IMPDH2 recues gene expression and alternative splicing. Intriguingly, previous work showed that MPA reduces SARS-CoV-2 replication(Gordon, Jang, et al. 2020). In summary, we believe that our model recapitulates some events of SARS-CoV-2 infections, although we are aware of its limitations. In addition, we think that the suggestion of looking into how different pairs of protein work together is a great idea (see below) although outside of the time frame of this work. In response to this idea, we have included a few lines in the discussion about this. We thank the reviewer for bringing up this point.

– Again, here, for Figure 5, it would be very important to know WHEN Nsp14 is actually expressed/present in the cell. There are not many changes after 8 hours, but how much Nsp14 have the 293 cells even made at this point?

We thank the reviewer to rise this important point. This is indeed key

because we observed the first changes that might be important to understand the cascade of events that lead to the effect of Nsp14. We addressed this question in two ways:

We performed WB at different timepoints and observed expression of Nsp14 from the 6-hour post transfection timepoint. These results are shown in Figure 5—figure supplement 1B. These results are consistent with our 3’RNAseq data showing the first genes differentially expressed upon Nsp14 transfection.In addition, we determine the normalized number of reads of *Nsp14* in our 3’ RNAseq dataset. The result is reported in Figure 5—figure supplement 1A and show robust expression of *Nsp14* mRNA, even at early timepoints.

The fact that "more than 50% of the changes in the transcriptome upon Nsp14 expression are transcriptional" seems like it warrants a bit of investigation and hopefully some imaging or subcellular fractionation at different time points could at least provide a framework to propose hypotheses for how this is possible in the context of the other mechanisms you're proposing

We thank the reviewer for bringing this up. We acknowledge that the statement was confusing. We based our assessment of how much of the changes are transcriptional on the amount of intronic signal of the differentially expressed genes in the total RNAseq, but we didn’t imply that Nsp14 acts directly to activate transcription but through the proposed pathways. Indeed, although the hypothesis that Nsp14 works as a transcriptional or nuclear factor would lead to a compelling model, we don’t think that the transcriptional effects are directly due to the binding of Nsp14 to chromatin or even its localization to the nucleus, and we provide now experiments to show that (subcellular fractionation and chromatin isolation at different timepoints showed in Figure 6—figure supplement 1A). Nevertheless, we verified that there is a transcriptional effect for a subset of genes that are deregulated upon Nsp14 expression by performing RT-qPCR from nascent RNA (showed in Figure 2F). We have also provided a much better explanation in the discussion of the possible mechanisms that could make this possible. We are grateful to the reviewer for bringing this up as it was really confusing before, and we feel that it is much clearer now.

– One could also do a chromatin-fractionation to see if Nsp14 sticks to chromatin, better / more complete to do ChIP-seq but this could be overkill for this study

Following this suggestion, we indeed performed subcellular fractionation and chromatin isolation and couldn’t find Nsp14 bound to chromatin or even in the nucleus. Regarding IMPDH2, previous studies showed that IMPDH2 can be found in the nucleus(Ahangari et al. 2021). Although we did not investigate this in more detail, we agree that ChIP might be a good way to test whether Nsp14 alters IMPDH2 localization and nuclear function in the future and have discussed this possibility in the discussion. However, all our evidence points towards a different mechanism of action of Nsp14.

The circRNA changes are interesting at a high level however it is hard to know how surprising or unique this change is; both that it changes and the magnitudes (number of circRNAs changing). Are there data that this could be compared to? For example, KD, KO, or OE of other splicing factors in experimental or disease conditions that show fewer, similar, or more changes in circRNAs, and importantly how these numbers compare to the changes in the linear mRNA pool.

We thank the reviewer for this keen question about circRNAs mis regulation. In our dataset we see about -250 circRNAs upregulated upon Nsp14 expression. This is indeed a large fraction, as we detected about 2,400 circRNAs in the control cells. Intriguingly, most of the upregulated circRNAs do not derive from genes that are also upregulated upon Nsp14 expression suggesting that this is specific regulation of exon circularization rather than the mere consequence of increased transcription from the locus (we now showed these results in Figure 3 and explained in the Result section “Nsp14 expression provokes changes in alternative splicing and circRNAs production”). We have now highlighted this information as it was clearly not well explained before. We hypothesize that accumulation of circRNAs could be linked to changes in the cell cycle or proliferation state of the cell (we have now expanded this on the Discussion). In addition, it is difficult to make a fair comparison between our experimental designs and previously published studies about circRNAs expression in the context of pathologies, but previously published examples of circRNAs altered in disease conditions or upon expression of specific splicing factors, generally report smaller numbers of circRNAs changing.

Below we show some examples from recent studies which show how different physiological contexts and genetic manipulation of factors involved in circRNAs biogenesis can result in altered expression of global circRNAs:

Bachmayr-Heyda and colleagues reported global reduction of circRNAs in colorectal cancer cell lines and cancer compared to normal tissues. They found 39 circRNAs (out of the predicted 1,812) being mis regulated (11 up and 28down). Furthermore, they discovered a negative correlation of global circRNA expression and cell proliferation, postulating a model about circRNAs accumulation in non-proliferating cells (Bachmayr-Heyda et al. 2015). This is particularly relevant to our case. Indeed, Nsp14 alters the expression of cell cycle controlling genes and this could potentially lead to cell cycle arrest. In the future we would like to assess the impact of Nsp14 on cell cycle and investigate the impact of cell cycle arrest on circRNAs accumulation.Conn and colleagues (Conn et al. 2015) showed that QKI is a crucial factor for circRNA biogenesis in the EMT context. Indeed, they reported that out of 300 abundant circRNAs in mesenchymal human mammary epithelial cells, 105 were decreased in QKI knockdown cells, while only 7 were increased.Errichielli and colleagues (Errichelli et al. 2017) showed that FUS is important for regulating circRNA expression in mouse embryonic stem cells (mESCs). In mESC derived from FUS-/- mice, they found 136 deregulated circRNAs, in comparison to cells derived from FUS +/+ mice. Remarkably, 111 out of the 136 circRNAs were downregulated.Nova 2 is another factor involved in circRNA regulation in the murine brain. Knupp and colleagues reported 78 downregulated and 43 upregulated circRNAs in Nova2-KO brain in respect to the control(Knupp et al. 2021).

We hope these examples may provide a better idea about the factors and the physiological contexts that induce circRNA deregulation, as well as of the magnitude of the effects. We mentioned some of these examples also in the Discussion.

The experimental idea in Figure 4, with co-expression is quite nice however limited in scope. This may be ok for this study, but it is hard to know how all these factors work together during a native infection when studying them in isolation. It's convenient that Nsp14 has such a singularly robust impact of the host transcriptome, but one wonders how the combination of the other gene products would better tune or in other ways influence the host cell's state.

We thank the reviewer for bringing this up. We do agree that it will be very interesting to perform additional co-expression experiments. Indeed, Nsp14 is part of the replication complex and therefore it works with several other viral proteins to provide proofreading activity. Pull down assays showed that Nsp14 interacts with nsp12-nsp7-nsp8 (replicase and its cofactor)(Romano et al. 2020; Subissi et al. 2014). Furthermore, it is also important for the modification of the 5’ cap of the viral RNA, and for this scope it works closely with other proteins involved in this process(Y. Chen et al. 2009). In this context, we agree that could be really informative to dissect the function of these interactions on the cell transcriptome. We have now added this to the discussion.

[Editors' note: further revisions were suggested prior to acceptance, as described below.]

Reviewer #1 (Recommendations for the authors):The authors addressed most of the points raised by reviewers and enhanced the manuscript with compelling data. Although they did not address the comment regarding the role of Nsp14 function in authentic infection, I think that the function of Nsp14 observed upon exogenous expression is well addressed. I do understand the limitation of doing BSL3 experiment. But here I have some questions and suggestions related to their response to this comment.1. The authors showed the cytoplasmic localization of exogenous Nsp14 as one of the supporting evidence of resemblance to SARS-COVID-2 infection situation. I wonder how cytoplasmic localization of Nsp14 could be associated with altered splicing events which occur in the nucleus. It would be nice to include any speculation on this in the discussion.

We thank the reviewer for pointing out this puzzling evidence. Based on our evidence, we claimed that Nsp14 does not directly regulate gene expression and alternative splicing in the nucleus but that cellular proteins mediate these effects. For example, we claim that IMPDH2 mediates both gene expression changes and alternative splicing, but we cannot exclude that also other proteins might be involved in the process. We changed the discussion to make this point clearer.

2. To further support the role of Nsp14 in altered splicing upon the real SARS-CoV-2 infection, I would suggest reinforcing the specificity of Nsp14 on altered splicing. Given that Nsp16 is also known for altered splicing, especially causing intron retention which is the common phenomenon the authors observed with Nsp14, I still wonder how much Nsp14 would explain the altered splicing upon SARS-COVID-2 infection.

We thank the reviewer for stressing this interesting point. Indeed, we are very interested in comparing the effect of Nsp14 and Nsp16 to determine which is the contribution of each of them to alternative splicing during SARS-CoV-2. We included in the Discussion possible ways to address this question with future experiments.

Reviewer #2 (Recommendations for the authors):The authors carefully read the initial reviews, extended their work, and I think addressed the major of the conceptual points raised. The additional data and controls included provide better context in some cases and strengthen the mechanistic claims overall. Parts of the news results and Discussion sections I think also better contextualize some of the initial ideas that were less clearly stated in the first draft. As far as I am concerned, I think this manuscript should proceed to publication.

We thank the reviewer for the positive feedback.

Furthermore, we would like to point to the editor and the reviewer’s attention that we modified the following minor points:

– We updated the acknowledgement section as we realized that we forgot to mention some contributors in the previous version of the manuscript

– We corrected the founding section

– We realized that we mentioned ORF3b in the manuscript. However, we found some studies that reported a mistake in the nomenclature of this protein/plasmid (Jungreis et al., Virology 2021 and Nelson et al., *eLife* 2020). The sequence cloned in the plasmid for our screening was ORF3d. We changed the figure and the text accordingly.